

# Importance of seasonally resolved oceanic emissions for bromoform delivery from the tropical Indian Ocean and west Pacific to the stratosphere

Alina Fiehn[1, 2,+], Birgit Quack[2], Irene Stemmler[3], Franziska Ziska[2,*], Kirstin Krüger[1]

[1] *Meteorology and Oceanography Section, Department of Geosciences, University of Oslo, Oslo, Norway*
[2] *GEOMAR Helmholtz Centre for Ocean Research Kiel, Kiel, Germany*
[3] *Max Planck Institute for Meteorology, Hamburg, Germany*
*+ now at: Deutsches Zentrum für Luft- und Raumfahrt, Oberpfaffenhofen, Germany*
*\* now at: Deutscher Wetterdienst, Hamburg, Germany*

Alina Fiehn: alina.fiehn@dlr.de
Birgit Quack: bquack@geomar.de
Irene Stemmler: irene.stemmler@mpimet.mpg.de
Franziska Ziska: franziska.ziska@dwd.de
Kirstin Krüger: kkrueger@geo.uio.no (corresponding author)



## Abstract

Oceanic very short-lived substances (VSLS), such as bromoform ($CHBr_3$), contribute to stratospheric halogen loading and, thus, to ozone depletion. However, the amount, timing, and region of bromine delivery to the stratosphere through one of the main entrance gates, the Asian monsoon circulation, are still uncertain. In this study, we created two bromoform emission inventories with monthly resolution for the tropical Indian Ocean and west Pacific based on new

in situ bromoform measurements and novel ocean biogeochemistry modeling. The mass transport and atmospheric mixing ratios of bromoform were modeled for the year 2014 with the particle dispersion model FLEXPART driven by ERA-Interim reanalysis. We compare results between two emission scenarios: (1) monthly and (2) annually averaged emissions. Both simulations reproduce the atmospheric distribution of bromoform from ship- and aircraft-based observations

in the boundary layer and upper troposphere above the Indian Ocean well.

Using monthly resolved emissions, main oceanic source regions for the stratosphere include the Arabian Sea and Bay of Bengal in boreal summer and the tropical west Pacific Ocean in boreal winter. The main stratospheric entrainment in boreal summer occurs over the southern tip of India associated with the high local oceanic sources and strong convection of the summer

monsoon. In boreal winter more bromoform is entrained over the west Pacific than over the Indian Ocean. The annually averaged stratospheric entrainment of bromoform is in the same range whether using monthly or annually averaged emissions in our Lagrangian calculations. However, monthly averaged emissions result in highest mixing ratios within the Asian monsoon anticyclone in boreal summer and above the central Indian Ocean in boreal winter, while

annually averaged emissions display a maximum above the west Indian Ocean in boreal spring. In the Asian summer monsoon anticyclone bromoform atmospheric mixing ratios vary up to 50% between using monthly and annually averaged oceanic emissions. Our results underline that the seasonal and regional stratospheric bromine entrainment from the tropical Indian Ocean and west Pacific critically depends on the seasonality and spatial distribution of the VSLS emissions.






## 1 Introduction

Halogenated very short-lived substances (VSLS) contribute to the stratospheric halogen burden, take part in ozone depletion and, thus, impact climate (Law et al., 2006). They are of oceanic origin and their transport to the stratosphere mainly depends on deep convection in the tropics.

The contribution of oceanic VSLS is estimated to be 10-40 % of the current ~20 ppt total stratospheric bromine (Dorf et al., 2006 and updates). Uncertainties result mainly from tropospheric degradation and removal, transport processes, and especially from the spatial and temporal variability of halogenated VSLS emissions (Carpenter et al., 2014; Hossaini et al., 2016). In this study, we focus on the influence of seasonal emission variations.

Bromoform ($CHBr_3$) is one of the largest contributors to bromine from VSLS ($Br_y^{VSLS}$) in the stratosphere (Hossaini et al., 2012) due to its large oceanic emissions (Quack and Wallace, 2003), moderate tropospheric lifetime of 15-17 days in the tropics (Carpenter et al., 2014), and because it contains three bromine atoms. The bromoform surface concentration in the ocean is spatially and temporally variable and depends on its chemical and biological production (Carpenter et al.,

1999; Quack and Wallace, 2003). Enhanced emissions coincide with biologically active equatorial and coastal upwelling regions (Quack et al., 2007) and the distribution of macro algae and anthropogenic sources along the coasts (Carpenter and Liss, 2000; Quack and Wallace, 2003). There are different approaches in creating global bromoform emission inventories. As bottom-up approach, emissions are extrapolated from marine and atmospheric observations in different

spatial resolutions (Quack and Wallace, 2003; Butler et al., 2007; Palmer and Reason, 2009; Ziska et al., 2013). The top-down approach uses chemistry transport and chemistry climate models to infer possible emission distributions that reproduce observed atmospheric abundances of VSLS (Warwick et al., 2006; Liang et al., 2010; Ordóñez et al., 2012). Recently, an ocean biogeochemical model simulated oceanic bromoform distributions and derived emissions

(Stemmler et al., 2015) based on a marine production module (Hense and Quack, 2009) and observational atmospheric data (Ziska et al., 2013).

Overall, large differences between bromoform emission inventories exist. The bottom-up inventories (Ziska et al., 2013; Stemmler et al., 2015) estimate lower global bromoform emissions than the top-down inventories (Warwick et al., 2006; Liang et al., 2010; Ordóñez et al.,

2012). The top-down emission inventories and the Ziska et al. (2013) inventory have been compared and evaluated by Hossaini et al. (2013). The observation based bromoform emissions



of Ziska et al. (2013) led to the best agreement with tropospheric measurements of atmospheric mixing ratios in the tropics. Some emission inventories represent climatological annual means (Warwick et al., 2006; Liang et al., 2010) and other inventories include a seasonality of emissions

(Ordóñez et al., 2012; Ziska et al., 2013; Stemmler et al., 2015).

In the atmosphere VSLS are defined as having a lifetime shorter than half a year (Law et al., 2006). They degrade through photolysis or reaction with the hydroxyl radical (OH) into soluble substances, which can then be washed out from the troposphere. Stratospheric delivery of VSLS is connected to fast and high-reaching convection and ascent of air masses through the

tropical tropopause layer (TTL) into the stratosphere (Gettelman et al., 2009), because their degradation occurs on similar timescales as the transport. The main regions of entrainment of tropospheric air masses into the stratosphere lie over the tropical west Pacific Ocean in boreal winter and the Indian monsoon region in boreal summer (Newell and Gould-Stewart, 1981).

The emissions of VSLS from the Pacific Ocean, their atmospheric mixing ratios, and

transport to the stratosphere has been measured and modeled in various studies (Tegtmeier et al., 2012; Tegtmeier et al., 2013; Hossaini et al., 2016; and observations listed therein), but the uncertainty of Indian Ocean emissions and their contribution to stratospheric bromine is still large (Liang et al., 2014). The Indian Ocean emissions could be quite high based on two oceanic measurement campaigns in the marginal seas (Yamamoto et al., 2001; Roy et al., 2011), as well

as extrapolations from other oceans (Ziska et al., 2013) and top-down source estimates (Liang et al., 2010). They have the potential to significantly contribute to stratospheric bromine (Liang et al., 2014; Hossaini et al., 2016). Based on first measurements of enhanced surface concentrations of bromoform and dibromomethane from the subtropical and tropical west Indian Ocean in 2014, Fiehn et al. (2017) calculated strong emissions and diagnosed stratospheric entrainment of these

two VSLS in the southeastern part of the Asian monsoon anticyclone in July and August 2014 with Lagrangian model calculations. VSLS tracers with different lifetimes revealed a strong seasonality in the transport strength from the tropical west Indian Ocean to the stratosphere, with maximum transport in boreal spring, when the main uplift occurs over this ocean basin (Fiehn et al., under review).

The atmospheric distribution and the delivery of bromoform to the stratosphere have been the topic of global chemistry transport and chemistry climate modeling studies. These studies used different approaches to constrain the input of VSLS from the ocean to the atmosphere: fixed uniform VSLS mixing ratios in the boundary layer (Hossaini et al., 2010; Hossaini et al., 2012;



Morgenstern et al., 2017) or in the upper troposphere (Aschmann et al., 2009; Aschmann et al.,

2011; Aschmann and Sinnhuber, 2013), prescribed emissions as homogeneous fields (Dvortsov et al., 1999; Nielsen, 2001) or according to one of the emission inventories described above (Warwick et al., 2006; Hossaini et al., 2013; Liang et al., 2014; Tegtmeier et al., 2015; Hossaini et al., 2016), or prescribed water concentrations to calculate emissions online (Lennartz et al., 2015). From this large set, only few studies considered seasonally varying surface water

concentrations or emissions in the models (Lennartz et al., 2015; Tegtmeier et al., 2015; Hossaini et al., 2016).

The seasonality of atmospheric mixing ratios is influenced by varying emissions as well as chemical degradation and transport processes. Liang et al. (2010) could reproduce the seasonality of atmospheric bromoform mixing ratios in the lower troposphere from available

aircraft observations using annually averaged emissions, concluding that the seasonality was mainly determined by chemical loss in the atmosphere and tropospheric transport. On the other hand, Lennartz et al. (2015) were not able to match the observed seasonality in atmospheric bromoform mixing ratios at ground-based stations, concluding that a seasonality in the bromoform sources was missing.

Furthermore, available model studies are in disagreement over the main stratospheric entrainment season and location for bromoform over Asia and the Indian Ocean. Liang et al. (2014) modeled the highest upper tropospheric mixing ratios above the Indian Ocean during boreal winter based on the constant and zonally homogenous emission estimate by Liang et al. (2010). In a multi-model intercomparison study of eleven chemistry transport and chemistry

climate models, Hossaini et al. (2016) used three different emission inventories for each model (Liang et al., 2010; Ordóñez et al., 2012; Ziska et al., 2013) of which only one (Ordóñez et al., 2012) was applied with seasonality. Overall, the models mainly agreed on the seasonality of volume mixing ratio (VMR) maxima at the tropical averaged cold point tropopause (CPT), but the absolute values varied within a factor of three. The locations of the VMR maxima at the CPT

above the tropical west Pacific in DJF were model consistent, but model differences in the strength of the Asian monsoon signature in JJA were high and strongly dependent on the parameterization of convection in the free troposphere and mixing in the boundary layer (Hossaini et al., 2016).

Until now, the influence of seasonally varying emissions on the stratospheric entrainment

of VSLS through the Asian monsoon has not been investigated. The combination of spatially and



temporally varying marine emissions and high resolution atmospheric transport will help to answer the question of where and when the main oceanic bromine delivery to the stratosphere occurs above Asia and the Indian Ocean.

In this study, we investigate the influence of seasonally varying bromoform emissions from the tropical Indian and west Pacific Ocean on the stratospheric entrainment of bromoform and its mixing ratios in the TTL modulated by the Asian monsoon circulation. Our research questions for this study are: What is the influence of seasonal bromoform emissions on stratospheric entrainment through the Asian Monsoon? Which are the main oceanic source and

stratospheric entrainment regions and seasons for bromoform above the Indian Ocean? What is the difference of bromoform entrainment to the stratosphere between using monthly and annually averaged emissions?

      In Sect. 2, we describe the bromoform emission scenarios that we applied in our transport simulations and the Lagrangian model set up. In Sect. 3, we present and discuss the model results.

Uncertainties in our studies are addressed in Sect. 4 and Sect. 5 contains the conclusions.

## 2     Data and Methods

### 2.1     Emission inventories

We created two bromoform emission inventories for the tropical Indian Ocean and the west

Pacific in 2014 from existing observation (Ziska et al., 2013) and model (Stemmler et al., 2015) inventories scaled with new in-situ measurements from the tropical-subtropical Indian Ocean (Fiehn et al., 2017). These were used in the Lagrangian dispersion model to determine the transport of bromoform from the tropical Indian Ocean to the stratosphere in 2014, the year of the only existing oceanic data set form the west Indian Ocean, obtained during the OASIS cruise on

RV Sonne in July and August 2014.

      The emission inventories are based on oceanic concentrations and atmospheric mixing ratios of bromoform as described below (Fig. 1). We calculated emission fields with a monthly resolution for 2014 using the parameterization of air-sea gas exchange by Nightingale et al. (2000), adapted to bromoform according to Quack and Wallace (2003). The air-sea flux is

obtained as the product of a transfer coefficient, which mainly depends on wind speed, and the gradient between the VSLS concentration in water and air. We use ERA-Interim 1˚ x 1˚ monthly means in 2014 for the physical parameters wind speed, sea level pressure and sea surface





temperature. The climatological annual mean sea surface salinity field was taken from the World Ocean Atlas 2009. The annual mean of the monthly mean emissions are used for the annually

averaged emission scenario. For this study we only consider air-sea fluxes from the tropical Indian Ocean and west Pacific (IO/WP), here defined as the region within 30˚N - 30˚S and 30˚E - 160˚E. This area is also used for the particle releases in the Lagrangian simulations (Sect. 2.2).

As a first inventory, the Ziska et al. (2013) climatology was updated with new oceanic and atmospheric measurements from the Halocarbons in the Ocean and Atmosphere (HalOcAt)

database. This emission inventory will be called *Ziska Updated* in the following (red lines in Fig. 1). The Ziska et al. (2013) climatology is an observation-based global air-sea flux estimate of bromoform, dibromomethane and methyl iodide, calculated from the in 2011 available oceanic and atmospheric surface concentrations within HalOcAt. The available surface data was classified as coastal, shelf or open ocean data. The open ocean data was further divided into

21 regions according to the physical and geochemical characteristics of ocean and atmosphere. Measurements were interpolated on a 1˚x1˚ grid and extrapolated within the regions using longitude and latitude regressions. The most relevant modification for this study is considering additional observations from the tropical and subtropical west Indian Ocean in July and August 2014 (Fiehn et al., 2017), which are the only measurements of bromocarbons in the tropical open

Indian Ocean. In the original Ziska climatology, emission values in the Indian Ocean were based on extrapolations from other ocean basins (Ziska et al., 2013). For this study, the global climatological mean fields of bromoform oceanic concentrations and atmospheric mixing ratios are updated using the ordinary least-squared (OLS) method from Ziska et al. (2013). In the IO/WP area the climatological mean oceanic surface concentrations are 7.8 pmol m$^{-2}$ h$^{-1}$ and the

mean atmospheric surface mixing ratio is 1.9 ppt (Fig. 1). The climatological fields were used together with physical data (i.e. wind speed etc. as described above) of higher frequency to calculate monthly mean emissions.

As a second emission inventory, modeled oceanic concentration fields of bromoform (Stemmler et al., 2015) were scaled with in-situ observations from the tropical and subtropical

Indian Ocean (Fiehn et al., 2017). This emission inventory will be called *Stemmler Scaled* in the following (blue lines in Fig. 1). Stemmler et al. (2015) used a global ocean general circulation model with a biogeochemistry model (MPIOM-HAMOCC, Ilyina et al., 2013) to simulate bromoform cycling in the ocean and to derive emissions to the atmosphere. They used the Ziska et al. (2013) surface atmospheric mixing ratio climatology as boundary conditions for their online





air-sea flux calculations. In general, the Stemmler et al. (2015) surface oceanic concentration and air-sea flux climatologies are lower than previously published estimates (see Table 3 in Fiehn et al. (2017) for comparison of tropical bromoform emissions). The low emissions might be partly caused by the use of the temporally constant atmospheric mixing ratios, which are not consistent with the state of the ocean and atmosphere and have been shown to decrease bromoform

emissions in a chemistry climate model setup (Lennartz et al., 2015). In this study, we scaled the oceanic bromoform concentrations of the Stemmler et al. (2015) "Dia" experiment. This experiment includes a spatio-temporally variable bromoform production rate, which leads to the most realistic bromoform emission distribution. The scaling of concentrations was done based on oceanic measurements from the OASIS cruise in the Indian Ocean (Fiehn et al., 2017) and the

TransBrom transit across the west Pacific Ocean (Krüger and Quack, 2013). For every measured bromoform concentration in the ocean surface during these two campaigns, the model concentration in the corresponding grid box during the respective months of the climatology was selected and a linear scaling factor was calculated to obtain the measured value. The average scaling factor of 3.48 was then homogeneously applied to the modeled sea surface bromoform

concentrations in the IO/WP release area of Stemmler et al. (2015). For our *Stemmler Scaled* emission inventory, we use monthly mean surface oceanic concentrations, which were on average between 6.3 and 6.8 pmol $m^{-2}$ $h^{-1}$ in the IO/WP, and a constant atmospheric bromoform mixing ratio of 1 ppt, which corresponds to the observed average atmospheric mixing ratio of bromoform from the OASIS (Fiehn et al., 2017) and TransBrom ship campaigns (Ziska et al., 2013), to

calculate the monthly varying emission fields (Fig. 1).



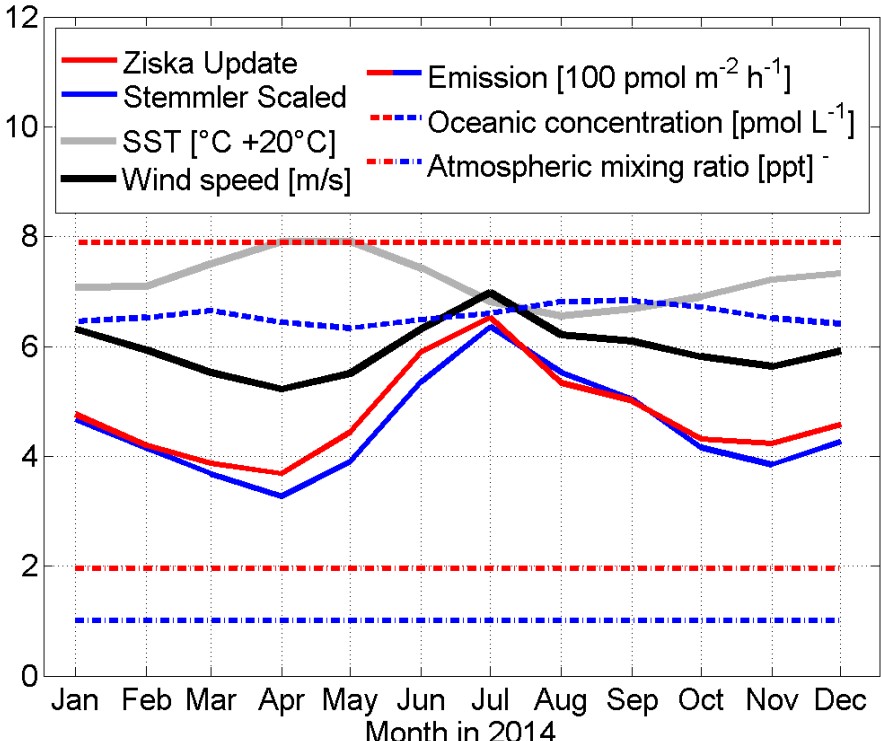

**Figure 1: Annual cycle of monthly mean emissions, surface water concentrations, and atmospheric mixing ratios of bromoform, as well as wind speed and SST, in the IO/WP release area in 2014 for the two inventories.**


## 2.2    FLEXPART calculations

For our transport calculations, we use the Lagrangian particle dispersion model FLEXPART of the Norwegian Institute for Air Research in the Atmosphere and Climate Department (Stohl et al., 2005), which has been evaluated in previous studies (Stohl et al., 1998; Stohl and Trickl, 1999).

The model includes moist convection and turbulence parameterizations in the atmospheric boundary layer and free troposphere (Stohl and Thomson, 1999; Forster et al., 2007). In this study, we employ version 9.2 of FLEXPART, which has been modified to incorporate atmospheric lifetime profiles for the decay of transported VSLS. We use the ECMWF reanalysis product ERA-Interim (Dee et al., 2011) with a horizontal resolution of 1° x 1° and 60 vertical

model levels as meteorological input fields, providing air temperature, winds, boundary layer height, specific humidity, as well as convective and large scale precipitation with a 3-hourly temporal resolution. The vertical winds in hybrid coordinates were calculated mass-consistently



from spectral data by the pre-processor (Stohl et al., 2005). We record the transport model output every 12 hours.

We ran the FLEXPART model using the monthly resolved and annually averaged emission fields for each of the Ziska Updated and Stemmler Scaled scenarios described above. According to these emission scenarios, we calculated the mass of bromoform released from each 1˚ x 1˚ grid cell during one day. We released one particle per day and grid cell from the IO/WP release area (30˚N - 30˚S, 30˚E - 160˚E) with the released mass attached. An exponential mass

decay is realized through the application of the lifetime profile of bromoform from Hossaini et al. (2010). For stratospheric entrainment we consider particles that reach above the CPT. We only calculate bromoform source gas injection to the stratosphere. The CPT is calculated online based on the ERA-Interim data. The seasonal mean CPT height is displayed in Fig. S1. We define the stratospheric *transport efficiency* as the mass of bromoform entrained to the stratosphere divided

by the emitted mass. FLEXPART output of trajectory positions and VMR fields is recorded 12-hourly and then averaged over one month.

### 3       Results

### 3.1      Bromoform emissions from the Indian Ocean/ West Pacific

The 2014 annual mean bromoform air-sea flux maps for the Ziska Updated and the Stemmler Scaled emission inventories in the IO/WP release area are shown in Figure 2. These emission distributions are used in the annually averaged emission scenario in FLEXPART. The Ziska Updated bromoform emission inventory includes high emissions along the northern hemispheric coastlines (1500-3000 pmol m$^{-2}$ h$^{-1}$) and in the central Bay of Bengal (up to 5000 pmol m$^{-2}$ h$^{-1}$).

An area of high emissions is the southern tropical Indian Ocean (1000 pmol m$^{-2}$ h$^{-1}$), while the flux from the tropical northwestern Pacific is negative, meaning that the ocean takes up bromoform from the atmosphere. This is caused by low oceanic concentrations, elevated atmospheric mixing ratios and low water temperatures, which enable the ocean to take up more gas from the atmosphere. The Stemmler Scaled bromoform emission inventory shows emission

hot spots at the Horn of Africa (2000 pmol m$^{-2}$ h$^{-1}$), south of the Oman coast (1700 pmol m$^{-2}$ h$^{-1}$), and in the Torres Strait north of the Cape York Peninsula of Australia (up to 5000 pmol m$^{-2}$ h$^{-1}$). The two inventories are similar in their main emission regions, the Arabian Sea and the Bay of Bengal, but show two differences: (1) Near the coast the Ziska Updated emissions are much





higher than the Stemmler Scaled emissions, because the bromoform module implemented into
HAMOCC (Stemmler et al., 2015) does not account for bromoform production by macro algae
and anthropogenic influences near the coastline. Furthermore, HAMOCC is a global carbon cycle
model not designed to represent coastal plankton growth/distributions, i.e. processes relevant on
shelves, such as riverine discharge of nutrients, tides or sediment resuspension are not considered.
(2) The air-sea fluxes in the west Pacific Ocean include negative fluxes north of 20°N in the
Ziska Updated inventory, while they are small but positive in the Stemmler Scaled inventory.

The seasonal mean emission fields show the intraannual variability of bromoform
emissions (Fig. 2). Emissions are high in boreal winter (December-February, DJF) and summer
(June-August, JJA) and lower in boreal spring (March-May, MAM) and fall (September-
November, SON) for both inventories. High emissions of the Ziska Updated inventory are
concentrated along the northern Indian Ocean coastline, the central Bay of Bengal and the
tropical southern Indian Ocean, with seasonal variations mainly driven by wind speed. Hot spots
in the Stemmler Scaled emissions mainly result from the high phytoplankton productivity in the
biogeochemical model (Stemmler et al., 2015).






**Figure 2:** Annual and seasonal mean bromoform emissions from the Indian Ocean and west Pacific release area (30˚N - 30˚S, 30˚E - 160˚E) of the inventories Ziska Updated (a) and Stemmler Scaled (b).




We compare the annual mean emissions of the two created emission inventories with their progenitors and two top-down inventories (Table 1). In Ziska Updated, the emission distribution

has changed compared to Ziska et al. (2013), while the Stemmler Scaled bromoform emission inventory mainly differs from Stemmler et al. (2015) in the total amount of bromoform emitted. The Ziska Updated inventory incorporates new high concentrations measured in the west Indian Ocean compared to Ziska et al. (2013), increasing the emissions in the southern Indian Ocean. Overall, the Ziska annual mean bromoform emission from the IO/WP increased from

670 Mmol Br yr$^{-1}$ (Ziska et al., 2013, OLS) to 750 Mmol Br yr$^{-1}$ (Ziska Updated, this study). The Stemmler Scaled emissions from the IO/WP release area are 760 Mmol Br yr$^{-1}$, while Stemmler et al. (2015) modeled only 43 Mmol Br yr$^{-1}$. The distribution has mainly remained the same for this inventory. The large difference in the emission strength results from scaling the surface water concentrations, the applied homogenous atmospheric mixing ratios of 1 ppt instead of the Ziska

et al. (2013) climatology and the ERA-Interim meteorological fields instead of the NCEP data used in Stemmler et al. (2015). The Ziska Updated and Stemmler Scaled inventories show similarities in the IO/WP region with previously published top-down bromoform emission inventories of Liang et al. (2010) and Ordóñez et al. (2012). Comparing spatial patterns, all six inventories in Table 1 include high emissions in the tropics, while only Liang et al. (2010)

assume zonally homogenous emissions. An estimate of the emission strength for coastal and open ocean regions of the Indian Ocean for the annual mean of six emission inventories is given in Table 1. The coastal emissions are similar for all inventories, except Stemmler et al. (2015), which is much lower, due to the lack of coastal macroalgal production and other potential processes relevant in the coastal ocean. The high emissions along the coast of Somalia and the

Oman in the Stemmler Scaled inventory are caused by high wind speeds during boreal summer and coastal upwelling, entailing bromoform production (Figure 2). This phenomenon is not captured in the Ziska Updated inventory due to missing bromoform measurements in this biogeochemical regime, but it is partly balanced by higher coastal emissions, like in the Liang and Ordoñez inventories.

The annual cycles of emission for Ziska Updated and Stemmler Scaled are very similar with a maximum in July and a secondary maximum in January and minima in April and November (Fig. 1). While the Ziska Updated inventory uses annually averaged oceanic concentrations, the Stemmler Scaled inventory includes monthly resolved concentrations calculated from temporally variable source-sink dynamics, such as production by plankton,

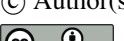



degradation, emissions, and transport by mixing and ocean currents (Stemmler et al., 2015). Both scenarios show the highest emissions in boreal summer, which has the highest wind speed, a quadratic factor in the air-sea gas exchange parameterization we used (Nightingale et al., 2000). We calculated the correlation between the annual cycles of emission from the IO/WP release area with each of the other variables (Table 2). Strongest correlations of the emission cycle exist with the wind speed and the SST. The correlation between emissions and oceanic concentrations for Stemmler Scaled are weak. Thus, from this table we can infer that the annual cycle of emissions in the Indian Ocean is mainly driven by the wind speed, which varies strongly over the year changing between the weak northeast (winter) and strong southwest (summer) monsoon winds.

**Table 1: Tropical coastal and open Indian Ocean (IO) annual mean emissions for different bromoform emission inventories. Values are given as fixed number, approximation, or range depending on the design of the inventory.**

| Inventory | Coastal IO emissions pmol m$^{-2}$ h$^{-1}$ | Open IO emissions pmol m$^{-2}$ h$^{-1}$ |
|---|---|---|
| Liang et al. (2010) | 1500 | 150 – 1100 |
| Ordóñez et al. (2012) | ~ 950 | ~350 |
| Stemmler et al. (2015) | 0 – 300 | 0 – 130 |
| Ziska et al. (2013) | 500 – 3000 | -300 – 800 |
| Ziska Updated, annual (this study) | 300 – 2500 | -300 – 800 |
| Stemmler Scaled, annual (this study) | 300 – 2300 | 200 – 800 |

**Table 2: Correlation between the annual cycle of bromoform emission and bromoform surface water concentration, wind speed and SST from Figure 1 using Spearman rank correlation. Bold face correlations are significant at 95% level according to a permutation test. Variables without an annual cycle (i.e. Ziska Updated surface water concentration and both atmospheric mixing ratios) could not be correlated.**

| Variable | Bromoform emission | |
|---|---|---|
| | Stemmler Scaled | Ziska Updated |
| Surface water concentration | 0.36 | - |
| Wind speed | **0.92** | **0.86** |
| SST | **-0.71** | -0.55 |




## 3.2    Comparison with observations

To evaluate our transport calculations, we compare the modeled VMR with observations from available ship and aircraft campaigns. A comparison of modeled and observed VMR may also determine where sources in the different emission inventories might be missing or have been

overestimated. We compare the modeled VMR at 1 km height with the ship cruise observations (Table 4, Fig. S2 and S3) from OASIS in the west Indian Ocean (Fiehn et al., 2017) and TransBrom across the west Pacific (Ziska et al., 2013). Note that we compare with the modeled VMR of the respective month of the cruise but always for the year 2014, and thus expect higher deviations for TransBrom (2009) than for OASIS (2014).

In the marine atmospheric boundary layer (MABL), which extends from the surface to about 1 km, high VMR mainly reflect emission hotspots. For both emission inventories, the modeled VMR at 1 km height are highest above the Indian Ocean and Asian coasts (Fig. S2). Beside their different coastal/open ocean distribution of emissions, they display similar hotspots in JJA and SON in the Arabian Sea and the Bay of Bengal, caused by high emissions in these

basins during the summer monsoon season with high surface wind speeds. For most cruise observations, the modeled VMR are lower than the measurements (Table 3, Fig. S3), which might be partly due to the experimental set up of including only emissions from the IO/WP. Likely oceanic sources from the tropical central Pacific also contribute to the VMR above the IO/WP (see Fig. 7 in Liang et al., 2014). Our comparison hints at missing coastal emissions in the

two inventories and an overall uncertainty in the tropical west Pacific emissions (Supplement text, Fig. S3).

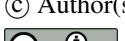



**Table 3: Mean bromoform volume mixing ratios (VMR, in ppt) observed during the research cruises and modeled with FLEXPART at 1 km height for the same location and month as the observation, but for the year 2014.**

| VMR [ppt] | Emission inventory | Emission scenario | West Pacific, TransBrom | Indian Ocean, OASIS |
|---|---|---|---|---|
| | | | Oct | July |
| Observation | | in situ | 0.92* | 1.28° |
| Modeled for 2014 | Ziska Updated | annually averaged | 0.11 | 0.72 |
| | | monthly averaged | 0.10 | 1.00 |
| | Stemmler Scaled | annually averaged | 0.32 | 0.43 |
| | | monthly averaged | 0.28 | 0.62 |

\* TransBrom 2009 (Ziska et al., 2013), ° OASIS 2014 (Fiehn et al., 2017)

In the free troposphere, we compare the modeled VMR with available observations from the CARIBIC aircraft observatory flights between November 2012 and February 2013 at around 11 km height above Southeast Asia (Wisher et al., 2014). The range and latitudinal gradient of FLEXPART VMR compare well with the aircraft measurements made between 15°N and 30°N (Fig. S4). Still, at the equator and in the north our simulations deliver less bromoform into the South Asian region than observed during CARIBIC. We account this to missing oceanic emissions from outside the release area in the central and east Pacific west of 160°E and to a likely underestimation of coastal sources in the two inventories. The assumption is supported by the fact that mixing ratios south of 10°N modeled with Ziska Updated are lower than those modeled with Stemmler Scaled (not shown) caused by lower emissions from the west Pacific in Ziska Updated.

The modeled annual mean mixing ratio profiles of bromoform up to 20 km height are largest over the Indian Ocean and lowest over the West Pacific (Figure 3a). Although mixing ratios are highest above the Indian Ocean, the above comparison with CARIBIC showed that also here contributions from the central Pacific may be important.



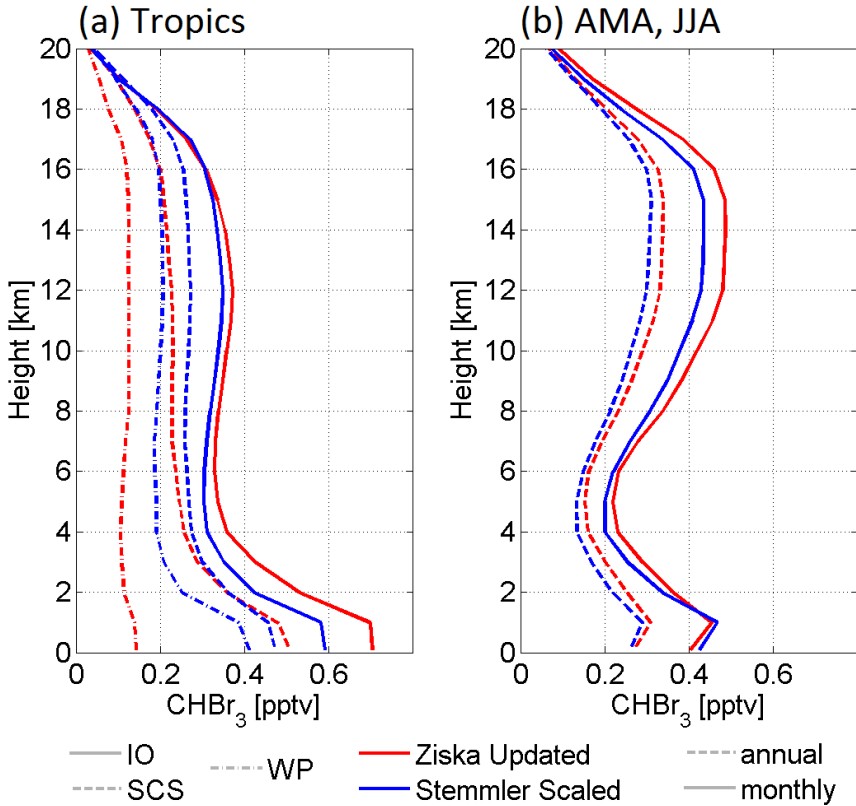

**Figure 3: (a) Annual mean tropical (20°S-20°N) VMR profiles of bromoform over the Indian Ocean (IO: 40°E-90°E), the South China Sea (SCS: 90°E-130°E), and the West Pacific (WP: 130°E-160°E) from the Ziska Updated and Stemmler Scaled inventories with monthly averaged emissions. (b) Bromoform VMR profiles in the Asian monsoon anticyclone region (AMA: 10°N-40°N, 20°E-90°E) in JJA from both inventories with monthly and annually averaged emissions.**

## 3.3    Ocean-to-stratosphere transport of bromoform

In this section we analyze the main oceanic source regions and stratospheric entrainment regions for bromoform from the Indian Ocean and the west Pacific. Here, we focus on emission scenarios with monthly variations. A comparison of stratospheric entrainment results using annually averaged emissions follows in the next section.

*Oceanic source regions*

Figure 4a and b show the distribution of bromoform mass delivered to the stratosphere displayed at the oceanic release locations, which depicts the oceanic source regions according to

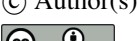



the two emission inventories. It also depicts the transport efficiency by source region (Fig. 4c) of bromoform delivery to the stratosphere, which results from dividing the delivered mass (Fig. 4a and b) by the air-sea fluxes (Fig. 2). The spatial distribution of transport efficiency by source

region is independent of the emission distribution and strength and the same for both emission inventories. The transport efficiency maps (Fig. 4c) show that the tropical west Pacific is the most efficient region at transporting bromoform from the ocean to the stratosphere in the annual mean. The maximum efficiency shifts from the WP equator in DJF toward the north in JJA and SON. In MAM the transport efficiency is more evenly distributed between the tropical Indian

Ocean and the west Pacific. In JJA the Bay of Bengal also displays elevated transport efficiencies.

        For both emission inventories (Fig. 4a and b) the most important source regions for bromoform to the stratosphere are the Asian coast and the Arabian Sea and Bay of Bengal, especially in JJA. During this season, the emissions from these regions are high (Fig. 1) providing bromoform to the Indian summer monsoon convection over India and the Bay of

Bengal. For the Ziska Updated inventory (Fig. 4a), the southern tropical Indian Ocean is also an important source region, while the open west Pacific delivers hardly any bromoform to the stratosphere during all seasons due to low and partly negative emissions. Using the Stemmler Scaled inventory (Fig. 4b), the equatorial west Pacific Ocean provides a secondary bromoform source to the stratosphere, which is strongest in DJF.



**Figure 4: (a) Oceanic source regions of bromoform delivered to the CPT for the Ziska Updated inventory with monthly averaged emissions. (b) Like (a) but for Stemmler Scaled. (c) Source region transport efficiency for bromoform delivery to the CPT from (a) and (b). The black box depicts the Indian Ocean/West Pacific release area.**


When comparing emissions (Fig. 2), delivered mass (Fig. 4a and b), and transport efficiency (Fig. 4c) we can determine the oceanic regions where the stratospheric delivery is determined by the emissions and those where the transport dominates stratospheric entrainment.





If source regions with high transport efficiency coincide with high oceanic emissions, the stratospheric entrainment from this region is mainly emission-driven; if source regions with high

transport efficiency coincide with low emissions, then the stratospheric delivery is transport-driven. Analyzing the annual mean, the Arabian Sea and the Bay of Bengal are emission-driven source regions for the stratosphere: These ocean basins show low transport efficiencies (2% - 5%), but due to the high emissions they deliver maximum bromoform to the stratosphere. The west Pacific is a transport-driven source region: It contributes to stratospheric delivery through

the generally high transport efficiency (6% - 9%) despite low emissions in this region in both the Ziska Updated and the Stemmler Scaled inventories. This also means that small changes in the VSLS emissions in the west Pacific will have a strong influence on the total mass delivered to the stratosphere, which makes it important to better constrain present and future emissions from this key region.

Tegtmeier et al. (2015) also identified important bromoform source regions in the tropical oceans. They used a combination of bromoform emissions from Ziska et al. (2013) and ODP calculations (Pisso et al., 2010) to infer the importance of different oceanic regions for stratospheric ozone depletion, including an emission seasonality. The tropical west Pacific significantly contributed to ODP-weighted emissions all year round. In accordance with our

results, the main contribution in boreal summer comes from the Asian coastal areas and the Indian Ocean.

*Stratospheric entrainment regions*

The modeled stratospheric entrainment regions for Ziska Updated and Stemmler Scaled

inventories with monthly averaged emissions are depicted in Figure 5. In the annual mean, the entrainment maximum for both inventories occurs over the southern tip of India (Fig. 5a/b, top row). This maximum results mainly from the strong emissions in JJA and the fast uplift with the Asian summer monsoon circulation. The Stemmler Scaled inventory also shows a secondary entrainment maximum over the equatorial west Pacific, which the Ziska Updated lacks. It is

present in all seasons, but most pronounced in DJF. The weaker stratospheric entrainment above the west Pacific from Ziska Updated is the most obvious pattern throughout all seasons and in the annual mean of the difference between the two inventories (Fig. 5c). The Ziska Updated




inventory, on the other hand, displays stronger entrainment above the Bay of Bengal, caused by the strong coastal and central-Bay of Bengal emissions in this inventory.


**Figure 5: CPT entrainment for monthly averaged bromoform emissions for (a) Ziska Updated and (b) Stemmler Scaled emission inventories. (c) Differences in entrainment between the two inventories (a) - (b). The black box depicts the Indian Ocean/West Pacific release area.**



### 3.4 Monthly vs. annually averaged bromoform emissions

For this study, we calculated the ocean to stratosphere transport of bromoform using monthly and annually averaged emission fields for 2014. This enables us to detect the differences between the two experimental set ups and to find out which season and region delivers most bromoform to the stratosphere above the Indian Ocean.

Figure 6 shows the annual cycle of bromoform emissions, transport efficiency, and entrainment comparing the monthly and annually averaged emission scenarios of Ziska Updated and Stemmler Scaled simulations plotted at the time of particle release from the ocean. The annual cycles of monthly averaged emissions display maxima in January and July and minimum emissions in April and November (Fig. 6a, see also Fig. 1). Monthly averaged emissions are higher than the annual mean from June to September. The annual cycle of bromoform transport efficiency to the stratosphere displays two maxima, one in July and one in January (Fig. 6b). Generally, the transport efficiency, calculated from the total IO/WP emissions and entrainment, is higher for the Stemmler Scaled than for the Ziska Updated emissions, because Stemmler Scaled has higher emissions in the west Pacific, which is the most efficient source region for the stratosphere (Fig. 4c). The combination of the emission cycles and transport efficiency results in the annual cycles of stratospheric entrainment (Fig. 6c). Using annually averaged emissions, the annual cycle of stratospheric entrainment has the same seasonality as the transport efficiency and a maximum from May to July. Using monthly averaged emissions, the annual cycle of entrainment is amplified due to the similar seasonality in emissions and transport efficiency. The very high emissions in JJA combined with highest transport efficiencies result in the highest stratospheric entrainment during this season using monthly averaged emissions. There is, thus, a temporal shift in the maximum entrainment season from MJJ using the annually averaged emission scenario to JJA applying the monthly resolution.

The total annual entrainment of bromoform to the stratosphere is similar for monthly or annually averaged emissions (Table 4), which shows that the emission seasonality (Fig. 6c) does not influence the total annual mass entrained to the stratosphere in our experimental set up for the year 2014. Nonetheless, the differing annual cycles of bromoform entrainment to the stratosphere for monthly and annually averaged emissions should also influence the regional pattern of entrainment to the stratosphere and, thus, the importance of different ocean basins for stratospheric delivery.



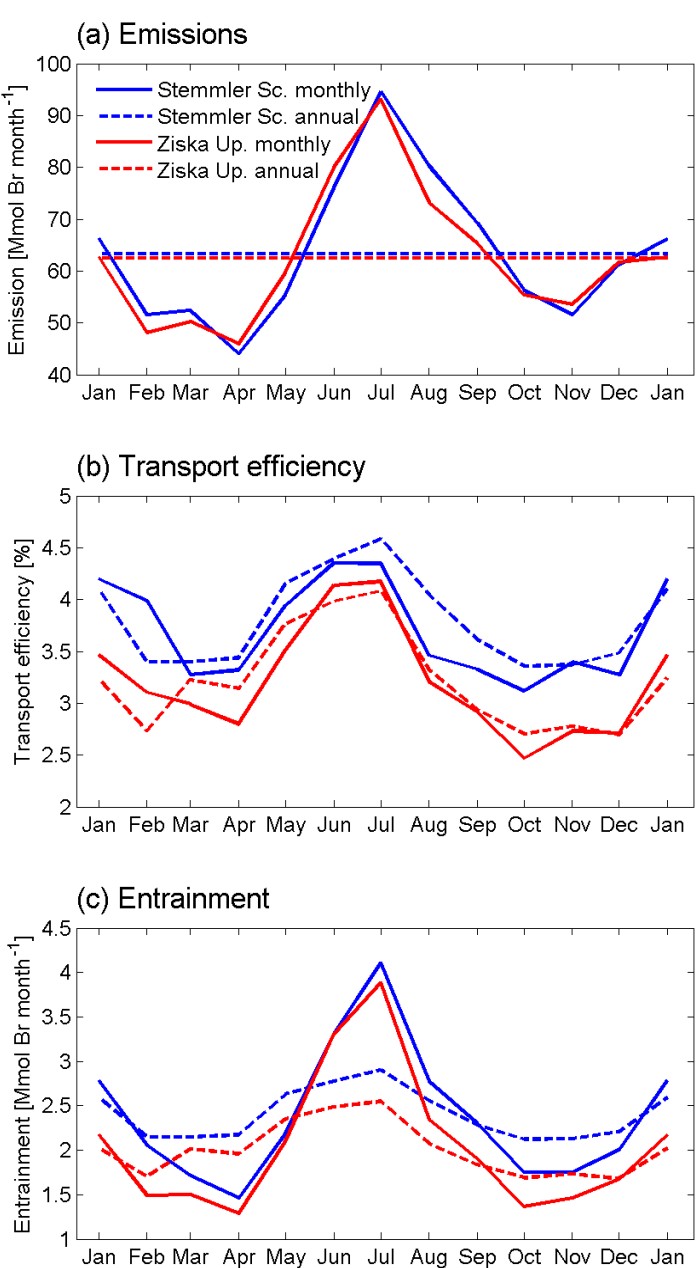

**Figure 6: Annual cycles of monthly sums of bromoform (a) emission, (b) transport efficiency, and (c) entrainment above the CPT in 2014 for the Ziska Updated and Stemmler Scaled monthly and annually averaged bromoform emission scenarios for the time of particle release.**





**Table 4: Total annual stratospheric entrainment of bromoform emitted from the IO/WP region.**

|  | Stemmler Scaled [Mmol Br yr$^{-1}$] | Ziska Updated [Mmol Br yr$^{-1}$] |
|---|---|---|
| Annually averaged emissions | 28.7 | 24.2 |
| Monthly averaged emissions | 28.2 | 24.5 |

We distinguish spatial differences in the stratospheric entrainment of bromoform between the monthly and annually averaged emission scenarios of the Ziska Updated inventory by examining the atmospheric VMR at 17 km altitude (Figure 7). This height is a good approximation for the tropical CPT height observed above the tropical Indian Ocean and West Pacific (Fiehn et al., 2017, not shown here), and it can be up to 18 km high in the Asian monsoon anticyclone in boreal summer (Munchak and Pan, 2014). High VMRs generally represent regions with enhanced uplift of bromoform from the ocean, but additionally indicate an accumulation in a certain region as for example the Asian summer monsoon anticyclone. Using annually averaged emissions (Fig. 7a), the maximum VMR region covers the tropical southwestern and central-northern Indian Ocean in the annual mean. This maximum is strongest for this scenario in MAM. For the monthly averaged emission scenario (Fig. 7b), the annual mean region of highest VMR at 17 km is also located above the tropical southwestern and central northern Indian Ocean, but the season with highest VMR is JJA. We, thus, diagnose different maximum bromoform VMR seasons using monthly vs. annually averaged emissions. The differences between VMR at 17 km for the two scenarios are displayed in Fig. 7c. In the annual mean, the VMR is lower north of 15ºN and higher south of 15ºN using annually averaged emissions than with monthly averaged emissions. In MAM, the annually averaged emissions deliver much more bromoform to 17 km height than monthly averaged emissions. This is reversed in JJA and SON, when monthly averaged emissions lead to higher VMR in the Asian monsoon anticyclone region in JJA and across the whole Indian Ocean/Asian area in SON. This difference in the VMR in the Asian monsoon anticyclone between the scenarios is also visible in the bromoform VMR profiles in Fig. 3b, revealing up to 50% more entrainment of bromoform into the UTLS region.





**Figure 7: Bromoform volume mixing ratios (VMR) at 17 km for the Ziska Updated (a) annually averaged emissions, (b) monthly averaged emissions, and (c) the difference between the two scenarios. The black box depicts the IO/WP release area.**



The respective figure for the Stemmler Scaled emission inventory displays very similar patterns (Fig. S5). The difference in the entrainment regions of bromoform mass at the CPT between the two scenarios also shows a similar seasonality in the anomalies between monthly and annually averaged emissions (Fig. S6), except for SON when the annual averaged emissions lead to higher entrainment but lower volume mixing ratios than monthly averaged emissions. This will be discussed in Sect. 4.

## 4 Discussion

This study investigates the influences of monthly vs. annually averaged bromoform emission representation in transport modeling above the tropical Indian Ocean and West Pacific and its stratospheric entrainment. We found seasonal and spatial differences in the VMR at 17 km between monthly and annually averaged emissions. They can be explained by the annual cycle of emissions and transport above the Indian Ocean. In DJF, the monthly averaged emissions are as high as the annually averaged emissions (Fig. 6a), causing only small differences in VMR at the tropopause (Fig. 7c). In MAM, monthly averaged emissions are lower than annually averaged emissions (Fig. 6a), causing lower VMR for monthly than for annually averaged emissions in the central Indian Ocean (Fig. 7c). In JJA, monthly averaged emissions reach their maximum, which is 50% (July) higher than the annually averaged emissions (Fig. 6a), and the transport efficiency through the Asian summer monsoon is also maximized (Fig. 6b), resulting in 50% (July) more bromoform entrainment with monthly averaged emissions than with annually averaged emissions (Fig 6c). Bromoform transported to the upper troposphere and lower stratosphere (UTLS) in boreal summer accumulates in the Asian monsoon anticyclone. The differing emission strength between annually and monthly averaged emissions causes a distinct signal in the difference of VMR at 17 km (Fig. 7c). In SON, the monthly averaged emissions are lower than the annually averaged emissions (Fig. 6a). However, we model higher VMR around the CPT with monthly than with annually averaged emissions, (Fig. 7c), which we interpret as a signal carried to SON from the previous season. The lifetime of bromoform in the TTL is around 25 to 30 days (Hossaini et al., 2010), which could be long enough for it to accumulate in the Asian monsoon anticyclone and then be distributed across the northern hemisphere over Asia in SON. This spreading of air masses from the anticyclone across the northern and also into the southern hemisphere during the breakup of the anticyclone in September has been observed for trace gases like CO, $H_2O$, and $O_3$ with satellites (Santee et al., 2017) and also simulated with a chemistry



transport model (e.g. Vogel et al., 2016). Thus, the negative anomalies in the anticyclone in JJA (Fig. 7c) influence the UTLS region in SON, while the sign of surface emission anomaly has already changed toward higher emissions from the annually than the monthly averaged emissions.

Regarding the annual mean VMR of bromoform at the tropopause, the representation of monthly resolved emissions results in a shift of bromoform toward the northern hemisphere UTLS, especially to the Asian monsoon anticyclone, because of the different main entrainment seasons: JJA using monthly averaged emissions and MAM using annually averaged emissions. Especially for comparisons with observational data, it is essential to consider the pronounced annual cycle of

VMR over the Indian Ocean.

The distinct difference in atmospheric bromoform mixing ratios in the Asian summer monsoon anticyclone between the annual and monthly averaged emissions is also visible in the VMR profiles of this circulation regime (Fig. 3b). At the level of main convective outflow (~14 km), using monthly averaged bromoform emissions results in 50% higher mixing ratios than

580 from annually averaged emissions. This large difference between the temporal distributions of emissions in a circulation regime with pronounced delivery to the stratosphere has a significant impact on stratospheric $Br_y^{VSLS}$ with 1.5 pptv versus 0.9 pptv source injections from bromoform monthly vs. annually averaged IO/WP emissions during the maximum entrainment season in boreal summer. Thus, we expect large seasonal differences for aircraft measurements of VSLS

above the ASM region (Fig. 7).

Our results help to interpret the discrepancy of modeled seasonality of bromoform VMR in the UTLS between Liang et al. (2014) and Hossaini et al. (2016). Both studies use global emissions of bromoform and show results only for annual emission scenarios. While Liang et al. (2014) simulate the VMR maximum for bromoform above the tropical Indian Ocean during DJF

using a chemistry climate model for 1960-2010, the set of chemistry climate and transport models for the period 1993-2012 from Hossaini et al. (2016) simulate the VMR maximum over the tropical West Pacific in DJF. They added that the contribution of the Asian monsoon pathway during JJA is highly uncertain, because of the large spread in the signal from model to model. The importance of DJF as season of enhanced stratospheric entrainment is connected with the

high transport efficiency above the tropical West Pacific during that season and, thus, also with the emissions of that region. The bromoform emissions inventory of Liang et al. (2010) used in Liang et al. (2014) has uniform high tropical emissions in the West and Central Pacific, which are transported towards the Indian Ocean and Asian continent (their Fig 7). The emission inventory



of Ziska et al. (2013), which is shown for some models in Hossaini et al. (2016), has low
emissions and even a sink in the northern tropical and subtropical West Pacific and high
emissions in the tropical Indian Ocean, resulting in a weak maximum of bromoform VMR above
the tropical West Pacific and Indian Ocean during JJA. We find in our study that the stratospheric
entrainment seasonality depends on the seasonal and regional distribution of bromoform
emissions from the IO/WP. Using seasonally varying oceanic bromoform emissions in our model
simulations increases the importance of the JJA entrainment to the stratosphere through the Asian
monsoon circulation.

## 5       Uncertainties

This study presents an estimate of bromoform entrainment to the stratosphere over the Indian
Ocean and Asia. Uncertainties in the analysis result from the emission inventories and the
FLEXPART model using ERA-Interim reanalysis fields.

      The Ziska et al. (2013) bromoform emission inventory was updated in this study by
including new observations. Available HalOcAt oceanic and atmospheric VSLS observations
contain a mixture of data from different seasons and years, which are used to calculate
concentration and mixing ratio climatologies. The seasonality in monthly averaged emissions
from Ziska Updated results only from the seasonality in wind speed, and sea surface pressure
used for the flux calculation. The Indian Ocean has a pronounced seasonality in ocean currents
and upwelling regions (Schott et al., 2009) affecting the biological productivity, surface
bromoform concentrations, and emissions (Quack et al., 2004; Hepach et al., 2015). Stemmler et
al. (2015) include seasonality in the modeled oceanic bromoform concentrations from
phytoplankton growth. This model study was designed to investigate processes that drive large
scale patterns of bromoform emissions from the open ocean and was carried out as climatological
steady state simulation. Thus, deviations from observations arise, for example, through missing
bromoform production from macroalgae along the coasts and unresolved temporal variability
patterns caused e.g. by ENSO (Stemmler et al., 2015). MPIOM-HAMOCC results derived with
the MPI-ESM were shown to be most realistic in the Indian Ocean compared to other CMIP5
models  (Roxy et al., 2016). Furthermore, our Stemmler Scaled inventory uses a temporally and
spatially uniform atmospheric bromoform mixing ratio to calculate the emissions. The annual
cycle of emissions results mainly from the changes in surface wind speed for both inventories



(Sect. 3.1). Thus, the annual cycle of oceanic concentrations plays a minor role in determining the annual emission cycle in the IO/WP region, but may become more important with higher resolution of the model and incorporation of coastal and open ocean emission hot spot regions. Furthermore, the parameterization for the air-sea flux itself is estimated to introduce an uncertainty of a factor of two (Lennartz et al., 2015).

The emissions and transport of VSLS in this study strongly depend on the ERA-Interim meteorological reanalysis and the boundary layer and convective parameterizations in the FLEXPART model. In most atmospheric models, convection, which occurs on scales smaller than the grid scale, is parameterized. The FLEXPART convection scheme was described and evaluated by Forster et al. (2007). FLEXPART ERA-Interim simulations have previously been

used to diagnose the VSLS transport and good agreement with aircraft measurements of bromoform, dibromomethane, and methyl iodide up to 13 km above the tropical West Pacific (Fuhlbrügge et al., 2016) and methyl iodide in the UTLS (Tegtmeier et al., 2013) was achieved.

If we want to infer the total delivery of $Br_y^{VSLS}$ to the stratosphere, we have to consider the oceanic source gases, but also the entrainment of their soluble product gases. Here, we only

consider the gases directly released from the ocean. The source gas injection into the stratosphere is generally enhanced with enhanced vertical uplift (Hossaini et al., 2010) and is overall estimated to contribute approximately half of the total stratospheric $Br_y^{VSLS}$ delivery of 2-8 pptv (Carpenter et al., 2014).

**6       Summary and Conclusions**

For this study, we compiled two new bromoform emission inventories for the tropical Indian Ocean and west Pacific (IO/WP) in 2014: An update (this study) of the Ziska et al. (2013) inventory including new measurements in the west Indian Ocean (Fiehn et al., 2017) and an inventory using monthly surface water concentrations modeled by Stemmler et al. (2015) and

scaled with measurements from the tropical Indian Ocean and west Pacific (this study). We calculated monthly emissions using climatological oceanic concentration for Ziska Updated and monthly oceanic concentrations for Stemmler Scaled and fixed annual mean atmospheric mixing ratios and SST combined with monthly mean wind speed and sea level pressure data. The resulting seasonality in bromoform emissions in the tropical IO/WP is mainly driven by wind

speed variations in the parameterized flux. The annual cycle of emissions for both inventories



displays maximum emissions during boreal summer located in the Bay of Bengal, Arabian Sea, and the tropical southern Indian Ocean.

We modeled the ocean-to-stratosphere transport for 2014 with FLEXPART based on ERA-Interim fields using monthly and annually averaged bromoform emission scenarios for both

inventories to detect the influence of seasonally varying emissions on stratospheric entrainment of VSLS. A comparison of modeled bromoform with observations from aircraft and ship observations from the Indian Ocean, the South China Sea, and the west Pacific displays that modeled mixing ratios were generally lower than observations due to our regionally restricted model set up and, thus, missing oceanic sources further upwind from the central and east Pacific

Ocean and possibly too low emissions along the coasts and in the northwest Pacific.

The oceanic source regions for stratospheric bromoform and the entrainment regions to the stratosphere for monthly averaged emissions were analyzed. For both emission inventories, most stratospheric bromoform originates from the Arabian Sea and Bay of Bengal in boreal summer and from the tropical west Pacific in boreal winter. The main annual mean entrainment

to the stratosphere occurs above the southern tip of India and results from the strong emissions from the Bay of Bengal and Arabian Sea and the efficient uplift with the Asian monsoon circulation during boreal summer.

We studied the influence of monthly resolved vs. annually averaged emission representation on the stratospheric entrainment and VMR of bromoform above the tropical

IO/WP region in 2014. We simulated similar total annual bromoform delivery to the stratosphere whether applying monthly or annually averaged emissions. However, monthly averaged emissions lead to less entrainment above the IO in boreal spring and more in boreal summer than annually averaged emissions. This causes up to 50% higher VMR in the Asian monsoon anticyclone and a change in the season with maximum VMR above the Indian Ocean at 17 km

height. Annually averaged emissions lead to highest VMR in MAM, while monthly averaged emissions cause highest VMR in JJA in the Asian monsoon anticyclone. The annual mean VMR at the tropopause using monthly averaged bromoform emissions are higher north of 15ºN and lower around the equatorial and in the southern hemisphere than with annually averaged emissions, probably caused by the enhanced stratospheric entrainment through the Asian summer

monsoon in the northern hemisphere.

Most surface-to-stratosphere transport above the Indian Ocean and Asia occurs in the Asian monsoon anticyclone during the summer monsoon and during its declining phase in boreal





fall. The use of temporally constant bromoform emissions, a common practice of many CTMs and CCMs (Warwick et al., 2006; Liang et al., 2014; Hossaini et al., 2016), significantly

influences stratospheric delivery seasonally and regionally. This contributes to the large uncertainty of modeled volume mixing ratios and stratospheric source gas delivery of bromoform and, thus, the stratospheric bromine loading (Liang et al., 2014; Hossaini et al., 2016). Although the modeled total annual bromoform delivery to the stratosphere does not vary much between monthly and annually averaged emission inventories in our 2014 IO/WP study, the region and

season of oceanic sources combined with effective atmospheric entrainment shifted the bromine pathway and seasonal and regional impact on the stratosphere. This is in particular of interest regarding future climate projections of stratospheric halogen loading with models projecting enhanced tropical deep convection (Hossaini et al., 2015) and a weakening Asian monsoon circulation (Christensen et al., 2013). This study was conducted for bromoform, but the impact of

representing seasonally resolved oceanic emissions for delivery from the Indian Ocean to the stratosphere also applies for other oceanic VSLS with lifetimes in the range and shorter than bromoform. We therefore strongly recommend using seasonally and regionally resolved oceanic VSLS emissions in chemistry transport and chemistry climate models for process studies or comparisons to observations, in particular for the tropical Indian and west Pacific Ocean and

Asian monsoon regions.





**Data availability**

The updated Ziska bromoform emission inventory data is available at Pangaea and the FLEXPART model output is stored at UNINETT Sigma 2 and can be inquired from the authors.

**Author contribution**

A. Fiehn, K. Krüger, B. Quack, and I. Stemmler designed the emission fields and the model experiments. A. Fiehn carried out the FLEXPART calculations and the model analysis. F. Ziska updated the Ziska et al. (2013) climatology for this study. A. Fiehn and K. Krüger wrote the manuscript with contributions from all co-authors.

**Competing interests**

The authors declare that they have no conflict of interest.

**Acknowledgements**

A. Fiehn was partly funded through the EU FP7 project StratoClim (603557). We thank the European Centre for Medium-Range Weather Forecasts (ECMWF) for the provision of ERA-725 Interim reanalysis data and the FLEXPART development team for the Lagrangian particle dispersion model used in this publication. The FLEXPART simulations were performed on resources provided by UNINETT Sigma2 - the National Infrastructure for High Performance Computing and Data Storage in Norway.




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
