# Peer review of "Importance of seasonally resolved oceanic emissions for bromoform delivery from the tropical Indian Ocean and west Pacific to the"

_Atmospheric Chemistry and Physics, 2018_

## Referee Comment (RC1) · M. Tao (Referee) · 3 Apr 2018

Review on **"Importance of seasonally resolved oceanic emissions for bromoform delivery from the tropical Indian Ocean and west Pacific to the stratosphere "**

General comments:

Based on two monthly resolved bromoform emission inventories and the atmospheric transport of bromoform modeled with the particle dispersion model FLEXPART, this study highlights that the seasonal and spatial variations of bromoform emissions are of importance for resolving the stratospheric entrainment, which contributes to the stratospheric halogen loading and ozone depletion. The paper presents important results. The approach and arguments are valid and the conclusions are reasonable. This manuscript is in good shape for publication in ACP. I recommend its publication after revision or answer towards the points below.

1. Overall, the introduction is comprehensive. However, some words about the importance of Asian monsoon in troposphere to stratospheric transport (TST) and previous related studies are necessary since this point is one of the main results. I suggest the author to cite some important works, especially some work about convective transport during ASM. Here are some recommendations for TST associated with ASM and its transport pathways. Overview:

   Randel, William J., et al. "Asian monsoon transport of pollution to the

   stratosphere." *Science* 328.5978 (2010): 611-613.

   convective transport:

   Orbe, Clara, Darryn W. Waugh, and Paul A. Newman. "Air-mass origin in the tropical

   lower stratosphere: The influence of Asian boundary layer air." *Geophysical Research*

   *Letters* 42.10 (2015): 4240-4248.

   Tissier, Ann-Sophie, and Bernard Legras. "Convective sources of trajectories traversing

   the tropical tropopause layer." *Atmospheric Chemistry and Physics* 16.5 (2016): 3383-

   3398.

2. Section 3.1, Line 278-285: another difference between the two inventories is also worth to mention: the hot spot in the central Bay of Bangle is pronounced the whole year in Ziska updated but not clear in Stemmler Scaled. This hot spot of emission is important for the delivered mass shown in Fig.4 (a). I also suggest to explain the formation of emission hot

spot.

3. Towards Figure 4, I am confused why the transport efficiency is independent of the emission distribution. According to the method (line 258-260),

Transport efficiency = $M_{strat.\ entrain}$/ $M_{emission}$

Thus, it should not be independent. Meanwhile, Fig. 6 (b) also shows different seasonality of transport efficiency from two inventories.

4. Figure 6 (a), why it show slightly different annual cycles from the red and blue solid lines in Fig.1? For example, the blue line is larger than the red in Jan.- Feb. but they are almost the same in Fig.1. Are these two figures showing the same quantity or not?

Specific comments:

1. what wind speed is used in Figure 1? Is it the monthly mean surface wind speed averaged in the IO/WP region? Please specify this either in the text or in the figure caption.

2. Page 10, line 265: … IO/WP release area are shown in **the top panel of** Figure 2.

3. Page 11, line 287-289: **Global mean** emissions are high….for both inventories **(see also Fig.1)**.

4. Figure 3, I recommend to use legends separately for each sub-figure, i.e. IO (Ziska updated) red solid line; IO (Stemmler Scaled) blue solid line and so on for (a).

5. Page 20, line 446: please specify ODP (ozone depletion potentials) since it is used for the first time.

6. Figure 5: please add statements that how the locations of the plotted quantity are decided.

---

## Referee Comment (RC2) · Anonymous Referee #2 · 6 Apr 2018

Review of "Importance of seasonally resolved oceanic emissions for bromoform delivery from the tropical Indian Ocean and west Pacific to the stratosphere", Fiehn et al.

General comments

This is an interesting paper that follows on from a series of similar studies by the same group investigating the sources of oceanic VSLS, their potential transport to the stratosphere and subsequent impact on ozone. The methods are mainly sound and have been reported previously so there are no major reasons that this paper cannot be published in ACP. Having said that there are a number of changes that I would like to see

before I can fully recommend the paper for publication.

The paper is reasonably well written but there are many instances where the clarity could be improved. Parts of the document are quite difficult to follow and could do with some revision. I have highlighted some of these in the specific comments below.

The section on comparison with available measurement data (Section 3.2) is a little weak for several reasons. Firstly, for the comparison with ship measurements, why did the authors choose model output at 1 km altitude when the ship is sampling much closer to the ocean surface? As is shown in, for example, Sala et al (ACP, 14, 2014), there can be a large gradient in VMRs between the surface and 1 km, which could easily account for the differences shown in Table 3 and Fig S3. Is there a reason why data from the 2011 SHIVA-Sonne cruise (South China Sea) was not included in the comparison? Similarly, for the aircraft comparison, there are a number of other recent campaigns in the region covered by the model which would have helped to further validate the flux and model/transport calculations. Examples include SHIVA (Sala et al. 2014), CAST (Andrews et al., AMT, 2016) and ATTREX (Navarro et al., PNAS, 112, 2015). When deriving new emission scenarios like this it is worthwhile testing the output against as much observational data as possible.

Many previous studies have discussed stratospheric entrainment/source regions in the tropics and I am not sure you have done sufficient justice to this previous work. Comparison of your findings with some of these other studies should be considered.

I struggle a little with the overall conclusion of this study. The two emission scenarios seem to produce quite similar results when looking at the region as a whole but are strikingly different when it comes to the actual fluxes from the ocean and the location of these fluxes (e.g. Figure 2). Can the authors begin to address which approach is more realistic/promising and perhaps discuss what the key areas that need further research are. How do we begin to reconcile the large differences between inventories? Do we simply need more observations?

Specific comments

L27-28 (also L93, L145, L154): be careful with the naming of the monsoon region. By Asian monsoon I presume you are referring to the Indian summer monsoon, rather than say the East Asian winter monsoon? Be consistent.

L35: I'm not convinced you can say that they "agree well" (there are significant differences in both the surface and upper troposphere comparisons). How about "agree reasonably well"?

L46: add "by", i.e. "vary by up to 50% ..."

L53: "they are of oceanic origin ...." Be specific – brominated VSLS are mainly oceanic but chlorinated VSLS are mainly anthropogenic.

L56: "Dorf et al and updates" – which updates are you referring to?

L56: "Uncertainties result from ..." I would argue that the uncertainty is also due to a lack of measurements of VSLS (both source and product gases) in the TTL and above.

L68-69: replace "As bottom-up approach ..." with "In the bottom-up approach ..."

L69-70: what is meant by "different spatial resolutions"? Do you mean ocean and atmospheric measurements in different locations?

L102: add "the" i.e. "Based on the first ...."

L102: "enhanced surface concentrations" – do you mean in the seawater or the atmosphere?

L108-109: should you add a date to the manuscript under review? Does the paper submitted to JGR differ significantly from this one (and Fiehn et al. 2017)?

L111: add "many" i.e. "... the topic of many global ...."

L119: change to "... only a few studies have considered ..."

L122-129: Is it worth discussing what factors might affect the seasonality in bromo-form sources here? What is the role of macroalgae relative to phytoplankton? The largest atmospheric concentrations are almost always near to exposed populations of seaweed. Do the emissions scenarios include this phenomenon? Annual changes in the tropics are presumably much less than at mid-latitudes and in polar regions?

L164-170: this section is a bit confusing. Are the new in-situ measurements (L166) from the OASIS cruise (L169)? L167: "These were used . . . ." What were used (the new inventories?) and where (in Fiehn 2017 or do you mean in this work?). This whole paragraph should be written more clearly.

L185: add "discussion" i.e. "in the following discussion"?

L187: move "in 2011" to the end of the sentence.

Figure 1 and Figure 3: it is difficult to distinguish between the different dashed lines. Can you try different line symbols?

Figure 1: Why are the atmospheric VMRs used in the 2 inventories so different? What would be the effect if both used the same atmospheric concentration? What impact does halving (or doubling) the atmospheric level have on the flux calculation?

L256-257: "We only calculate bromoform source gas injection to the stratosphere". Do you mean that you do not consider product gases at all? Perhaps you should state this for clarity?

L268: what drives the high emissions along the NH coastlines? Macroalgae? I guess you imply this later on (L 280-281) but why not state it here first?

L272: what is meant by "elevated atmospheric mixing ratios"? Where would the elevated levels come from?

L278: add "significant", i.e. "but show two significant differences. . ."

L279-283: If the Stemmler approach does not consider macroalgae and the effect of

coastal processes then surely it will always underestimate bromoform emissions? How important are macroalgae relative to phytoplankton, particularly in these regions?

L287-289: It is not clear from Figure 2 that emissions are necessarily higher in winter and summer than they are in spring and fall. Can a more robust or statistical case be made (total flux from the region in each of the 4 seasons for example)?

Figure 2: What is the cause of the high winter emissions (in the Ziska inventory) from the Chinese and Vietnamese (and Philippines?) coastlines? This appears to be a strong source region that you do not really discuss in the text. Given the prevailing NE winds at that time of year this could be an important source of bromoform to the tropics (see, for example, Ashfold et al., ACP. 15, 2015 or Oram et al., ACP, 17,2017)

Line 317: I agree that the coastal emissions are similar in magnitude but they are vastly different in location.

Table 1: If the numbers given are annual averages, what do the ranges shown represent?

Table 1: Are these numbers just for the Indian Ocean (i.e. not the full geographical area shown in Fig 2)? Please define what is meant by the Indian Ocean. Also I wonder if you should avoid using the term IO as it could be mistaken for iodine oxide!

L352: add "flux", i.e. "To evaluate our flux and transport calculations"?

L352-353 replace "available" with "selected", i.e. "from selected ship and aircraft campaigns".

L356: should be "Table 3" not "Table 4"

L368: Begin sentence with "It is likely that oceanic sources......"? Although please refer also to my general comments on Section 3.2 above.

L382-384: another, and possibly more likely (?), explanation would be the underestimation of the role of convection in this region. How well does FLEXPART deal with

convection?

L405 —-: In this section I think you should describe Figs 4a and 4b before discussing Fig 4c. As written, it is a little confusing.

L407-409: I am slightly confused by the term "transport efficiency" and how this was derived. In lines 258-260 it was defined slightly differently than it is here. How is the spatial distribution of transport efficiency independent of the emission scenario used when the mass emitted is different for the 2 scenarios? As I understand it, Fig 4c is a general picture which shows from which regions idealised particles will cross the CPT and has nothing to do with the bromoform emission inventories at all? If I am right, the term "bromoform delivery" in the Figure caption is misleading. A little clarification here would be appreciated.

Figure 4c. I am intrigued as to how the particles in the north east corner of the map get into the stratosphere during the summer months (JJA) when the prevailing winds in the region are from the southwest. Do they enter through the Indian monsoon or by some other mechanism?

L450: "Asian coastal areas" is a bit general. Which bit of Asia?

L454: I think you need to define again what you mean by the "stratospheric entrainment region". Please explain clearly what is depicted in Figure 5 and how it differs from Figure 4. Does Figure 5 show the geographical location at the CPT where particles pass through to the stratosphere? If so, it seems odd that the southern tip of India is so important when I thought the main convection occurs further to the north?

L470 (Section 3.4): this section would benefit from a better description of the difference between transport efficiency and entrainment (as discussed above).

L491-493: the temporal shift is not particularly obvious from Fig 6c.

L497-500: This sentence is not very clear. How do the "differing annual cycles of bromoform entrainment to the stratosphere" influence the "regional pattern of entrainment

to the stratosphere"?

L552 and L553: exactly 50% higher or approximately 50% higher?

L556: What altitude range does the anticyclone typically cover?

L615-617: "seasonality is only affected by wind speed and ss pressure". Is that because the atmospheric and ocean concentrations are assumed to be constant throughout the year?

L617-619: "The Indian Ocean has a pronounced seasonality in ocean currents and upwelling regions (Schott et al., 2009) affecting the biological productivity, surface bromoform concentrations, and emissions". Why include this sentence here? Do you mean to say that these are not included in the Ziska calculations? If so, please say so for clarity.

L645-649: This last sentence is not clear. What contributes "approximately half of the total stratospheric VSLS-Br"? Source gases in general? Where does the other 50% come from – product gases?
* * *

---

## Author Comment (AC1) · 20 Jun 2018

**Answer to the reviewers**

We would like to thank the two reviewers for the suggestions to improve the manuscript. Below you find our answers to their comments. The reviewer's comments are written in normal font, our answers in italics.

Additionally, we decided to change the word 'entrainment' to 'injection' as this is the more common term for trace gas transport from the troposphere into the stratosphere.

**Reviewer 1:**

**General comments**

1. Overall, the introduction is comprehensive. However, some words about the importance of Asian monsoon in troposphere to stratospheric transport (TST) and previous related studies are necessary since this point is one of the main results. I suggest the author to cite some important works, especially some work about convective transport during ASM. Here are some recommendations for TST associated with ASM and its transport pathways.
   Overview: Randel, William J., et al. "Asian monsoon transport of pollution to the stratosphere." *Science* 328.5978 (2010): 611-613.
   Convective transport: Orbe, Clara, Darryn W. Waugh, and Paul A. Newman. "Air-mass origin in the tropical lower stratosphere: The influence of Asian boundary layer air." *Geophysical Research Letters* 42.10 (2015): 4240-4248.
   Tissier, Ann-Sophie, and Bernard Legras. "Convective sources of trajectories traversing the tropical tropopause layer." *Atmospheric Chemistry and Physics* 16.5 (2016): 3383-3398.

   *We added an introduction of Asian monsoon transport in line 93.*
   *"Especially the Indian summer monsoon has been shown to transport boundary layer air masses into the stratosphere (Randel et al., 2010). Vogel et al. (2015) investigated the source regions and the dynamics of the Asian monsoon anticyclone, which strongly influences the transport in the Asian upper troposphere and lower stratosphere (UTLS) during boreal summer. While Orbe et al. (2015) researched the influence of Asian boundary layer air in the anticyclone, Tissier and Legras (2016) detected convective sources of air masses crossing the tropopause in this region. Recently, measurements of atmospheric trace gases in the anticyclone showed both stratospheric and boundary layer influences within the Asian monsoon anticyclone (Gottschaldt et al., 2017)."*

2. Section 3.1, Line 278-285: another difference between the two inventories is also worth to mention: the hot spot in the central Bay of Bengal is pronounced the whole year in Ziska Updated but not clear in Stemmler Scaled. This hot spot of emission is important for the delivered mass shown in Fig.4 (a). I also suggest to explain the formation of emission hot spot.

   *The Ziska emission inventory hot spot in the Bay of Bengal results from a measurement campaign during January-March 1995 (Yamamoto et al., 2001). Since these are the only data from the highly undersampled region it is quite uncertain how temporally and spatially resolved emissions from the Bay of Bengal really are. But we agree that this region is a very*

*important source region for the stratosphere (Fig. 4) and, thus, worth to be discussed. We added the following sentence to the discussion Sect. 5: "Thus, deviations from observations arise, for example, through missing bromoform production from macroalgae along the coasts, fixed phytoplankton production rates, and unresolved temporal variability patterns caused e.g. by ENSO (Stemmler et al., 2015). Further differences in the spatial emission distribution between the two inventories result from limited available data in the Ziska climatology and from lacking sources and process understanding in the Stemmler climatology. One example is the data based emission hot spot in the Bay of Bengal, which is not existent in the Stemmler inventory."*

3. Towards Figure 4, I am confused why the transport efficiency is independent of the emission distribution. According to the method (line 258-260),
   Transport efficiency = Mstrat. entrain/ Memission
   Thus, it should not be independent. Meanwhile, Fig. 6 (b) also shows different seasonality of transport efficiency from two inventories.

   *Correct, the transport efficiency is not independent of the emission distribution. But since it is very similar for both emission inventories we first suspected this. The similarity is caused by the very similar annual cycles of the emission inventories. But as you have noted the annual cycle as well as the distribution are slightly different. We changed this in line 423: "The spatial distribution of transport efficiency is very similar for both emission inventories why we only show the distribution for Ziska Updated."*

4. Figure 6 (a), why it show slightly different annual cycles from the red and blue solid lines in Fig.1? For example, the blue line is larger than the red in Jan. - Feb. but they are almost the same in Fig.1. Are these two figures showing the same quantity or not?

   *These figures are not showing the same quantities. Figure 1 displays the emissions in the unit pmol $m^{-2}$ $h^{-1}$. From this flux we calculated the total mass emitted from this region during one month, which is displayed in Figure 6. The annual cycles of these two quantities are different due to the varying distribution of emissions over the year and the influence of the surface area of each grid cell, which decreases with larger distance from the equator.*

Specific comments:

1. What wind speed is used in Figure 1? Is it the monthly mean surface wind speed averaged in the IO/WP region? Please specify this either in the text or in the figure caption.
   *Yes, this is correct. We added it to the figure caption.*

2. Page 10, line 265: … IO/WP release area is shown in **the top panel of** Figure 2.
   *Correct. This has been changed.*

3. Page 11, line 287-289: **Global mean** emissions are high….for both inventories **(see also Fig.1)**.

*We changed it to "Emissions in the IO/WP area are high…" because we do not investigate global emissions here.*

4. Figure 3, I recommend to use legends separately for each sub-figure, i.e. IO (Ziska updated) red solid line; IO (Stemmler Scaled) blue solid line and so on for (a).
*We have changed the legend on Fig. 3 accordingly.*

5. Page 20, line 446: please specify ODP (ozone depletion potentials) since it is used for the first time.
*Done.*

6. Figure 5: please add statements on how the locations of the plotted quantity are decided.

*We changed the beginning of the figure caption to: "Amount of CHBr3 injected to the stratosphere within 1°x1° grid cells plotted on the geographical location of CPT crossing …"*

**Reviewer 2:**

**General comments:**

This is an interesting paper that follows on from a series of similar studies by the same group investigating the sources of oceanic VSLS, their potential transport to the stratosphere and subsequent impact on ozone. The methods are mainly sound and have been reported previously so there are no major reasons that this paper cannot be published in ACP. Having said that, there are a number of changes that I would like to see before I can fully recommend the paper for publication. The paper is reasonably well written but there are many instances where the clarity could be improved. Parts of the document are quite difficult to follow and could do with some revision. I have highlighted some of these in the specific comments below.

The section on comparison with available measurement data (Section 3.2) is a little weak for several reasons. Firstly, for the comparison with ship measurements, why did the authors choose model output at 1 km altitude when the ship is sampling much closer to the ocean surface? As is shown in, for example, Sala et al (ACP, 14, 2014), there can be a large gradient in VMRs between the surface and 1 km, which could easily account for the differences shown in Table 3 and Fig S3. Is there a reason why data from the 2011 SHIVA-Sonne cruise (South China Sea) was not included in the comparison? Similarly, for the aircraft comparison, there are a number of other recent campaigns in the region covered by the model which would have helped to further validate the flux and model/transport calculations. Examples include SHIVA (Sala et al. 2014), CAST (Andrews et al., AMT, 2016) and ATTREX (Navarro et al., PNAS, 112, 2015). When deriving new emission scenarios like this it is worthwhile testing the output against as much observational data as possible.

*The comparison with ship measurements has been changed to a comparison with model output from 100 m above sea level. The Figures S2 and S3 in the supplement were changed and the text in Sect 3.2*

*and Table 3 was adapted. The SHIVA comparison (Fig. R1) was added to the supplement and the averages were added to Table 4 in the main manuscript. Corresponding text was adapted to these figure changes.*

[Figure]

**Figure R1: Comparison of modeled bromoform volume mixing ratios (VMR) at 100 m height and ship cruise measurements in the (a) Indian Ocean in July 2014 (OASIS), (b) the west Pacific in October 2009 (TransBrom), and (c) South China and Sulu Sea in November 2011 (SHIVA).**

*A comparison of the model results with further aircraft campaigns in the West Pacific, such as SHIVA, CAST, ATTREX aircraft campaigns, is not suitable for this paper. We release CHBr3 in the IO/WP region and not globally. Due to the mainly westerly atmospheric circulation above the Pacific, this region has systematically lower atmospheric mixing ratio in our simulation, than observed. Right now, we have another paper in preparation by Tegtmeier et al. where we exactly carried out the FLEXPART model aircraft comparison using global VSLS emission by Ziska et al. (2013).*

Many previous studies have discussed stratospheric entrainment/source regions in the tropics and I am not sure you have done sufficient justice to this previous work. Comparison of your findings with some of these other studies should be considered.

*We added a section in the introduction on stratospheric injection through the Asian summer monsoon and its source regions as also suggested by Reviewer 1. However, a comparison of our Indian Ocean bromoform emission driven results with those introduced Asian summer monsoon studies is difficult, because we use spatial and temporal varying sources and lifetime profiles for bromoform in contrast to the other pure air mass transport studies (Chen et al., 2012; Bergman et al., 2013; Orbe et al., 2015; Vogel et al., 2015) or studies investigating long-lived trace gases like water vapor and carbon monoxide (James et al., 2008; Park et al., 2009; Ploeger et al., 2012; Yan and Bian, 2015; Pan et al., 2016). In our companion studies on VSLS transport from the Indian Ocean to the stratosphere (Fiehn et al., 2017; Fiehn et al., 2018), we added more discussion of previous available work.*

*Here in this study, we can compare with studies investigating Anticyclone composition or dynamics (Vogel et al., 2016; Gottschaldt et al., 2017; Santee et al., 2017) and especially with other VSLS studies (Liang et al., 2010; Hossaini et al., 2012; Liang et al., 2014; Hossaini et al., 2016) where a discussion has already been included in the manuscript in Section 4. It is hard to compare with VSLS studies that do not explicitly focus on the Asian monsoon area (Russo et al., 2015; Wales et al., 2018).*

I struggle a little with the overall conclusion of this study. The two emission scenarios seem to produce quite similar results when looking at the region as a whole but are strikingly different when it comes to the actual fluxes from the ocean and the location of these fluxes (e.g. Figure 2). Can the authors begin to address which approach is more realistic/promising and perhaps discuss what the key areas that need further research are. How do we begin to reconcile the large differences between inventories? Do we simply need more observations?

*We write in L. 370: "Our comparison hints at missing coastal emissions in the two inventories and reveals an overall uncertainty in the tropical west Pacific emissions (Supplement text, Fig. S3)." These are key uncertainties for our study and region investigated, but other uncertainties (Sect 5.) apply to this method as well. The number of oceanic VSLS measurements is very low, and demand a strong increase of the temporal and spatial data coverage. Future research directions also need to address direct Eddy covariance flux measurements of bromoform and other VSLS to reduce the uncertainties in the flux parameterizations.*

**Specific comments:**
L27-28 (also L93, L145, L154): be careful with the naming of the monsoon region. By Asian monsoon I presume you are referring to the Indian summer monsoon, rather than say the East Asian winter monsoon? Be consistent.
    *Thank you for this comment; I changed these cases to "Indian summer monsoon".*

L35: I'm not convinced you can say that they "agree well" (there are significant differences in both the surface and upper troposphere comparisons). How about "agree reasonably well"?
    *We added "reasonably" here.*

L46: add "by", i.e. "vary by up to 50% . . ."
    *Done.*

L53: "they are of oceanic origin ..." Be specific – brominated VSLS are mainly oceanic but chlorinated VSLS are mainly anthropogenic.
    *We changed the sentence to: "Brominated VSLS are mainly of oceanic origin and…"*

L56: "Dorf et al and updates" – which updates are you referring to?
    *We were referring to the newer WMO Ozone reports from Montzka et al. (2010) and Carpenter et al. (2014), which contain updates of the graphic from Dorf et al. (2006). We added these references in the manuscript.*

L56: "Uncertainties result from . . ." I would argue that the uncertainty is also due to a lack of measurements of VSLS (both source and product gases) in the TTL and above.
    *We added this: "…from a lack of VSLS measurements in the tropical tropopause layer (TTL)…"*

L68-69: replace "As bottom-up approach . . ." with "In the bottom-up approach ..."
    *Done.*

L69-70: what is meant by "different spatial resolutions"? Do you mean ocean and atmospheric measurements in different locations?

*Yes, we changed this to "locations".*

L102: add "the" i.e. "Based on the first ..."

*Done.*

L102: "enhanced surface concentrations" – do you mean in the seawater or the atmosphere?

*In the seawater; we added this.*

L108-109: should you add a date to the manuscript under review? Does the paper submitted to JGR differ significantly from this one (and Fiehn et al. 2017)?

*The JGR Paper (Fiehn et al., 2018) has been published now and the complete reference is added. All three publications investigate the same topic, but with different foci. The first concentrates on the IO VSLS measurements and the Asian summer monsoon transport, while the second elaborates on variability in transport during all seasons and over 15 years. Finally, this last publication combines bromoform emission and transport variability to achieve a better understanding of the combined processes. All three publications were combined in the PhD thesis of Alina Fiehn in 2017 entitled: "Transport of very short-lived substances from the Indian Ocean to the stratosphere through the Asian monsoon" delivered at the University of Kiel.*

L111: add "many" i.e. ". . . the topic of many global ..."

*Done.*

L119: change to ". . . only a few studies have considered . . ."

*Done.*

L122-129: Is it worth discussing what factors might affect the seasonality in bromoform sources here? What is the role of macro algae relative to phytoplankton? The largest atmospheric concentrations are almost always near to exposed populations of seaweed. Do the emissions scenarios include this phenomenon? Annual changes in the tropics are presumably much less than at mid-latitudes and in polar regions?

*We added the following sentences: "The Stemmler et al. (2015) emission inventory does not include effects of macro algae or other coastal sources, other than phytoplankton production. Bromoform production is simulated as a function of phytoplankton growth and is only applicable to the open ocean. Bromoform production in line with primary production shows a much less pronounced seasonal cycle in the tropics as compared to extratropical oceans, such as the Southern Ocean or North Atlantic. The seasonality in the Ziska et al. (2013) emissions is clearly driven by the winds."*

*A scaling between macro algal and phytoplankton emissions would need much more data and process understanding, which should be addressed in future research.*

L164-170: this section is a bit confusing. Are the new in-situ measurements (L166) from the OASIS cruise (L169)? L167: "These were used ..." What were used (the new inventories?) and where (in Fiehn 2017 or do you mean in this work?). This whole paragraph should be written more clearly.

*We restructured the paragraph to clarify.*

L185: add "discussion" i.e. "in the following discussion"?

*Done.*

L187: move "in 2011" to the end of the sentence.

*Done.*

Figure 1 and Figure 3: it is difficult to distinguish between the different dashed lines. Can you try different line symbols?

*This has been changed.*

Figure 1: Why are the atmospheric VMRs used in the 2 inventories so different? What would be the effect if both used the same atmospheric concentration? What impact does halving (or doubling) the atmospheric levels have on the flux calculation?

*The differences in the atmospheric mixing ratios of the two inventories result from the data chosen for the interpolation of the mixing ratio fields and the interpolation method. While the atmospheric mixing ratios from Ziska Updated undergo the whole process of division into different regions and interpolation on the grid, the atmospheric mixing ratios, we used to derive the Stemmler Scaled inventory, are distributed homogeneously in the release area. We tested using the Ziska Updated atmospheric mixing ratios with the Stemmler Scaled oceanic concentrations to calculate fluxes, but this resulted in unrealistic negative fluxes (into the ocean) along the coasts, as Stemmler is not addressing the coastal sources. Halving the atmospheric mixing ratios would result in an increase of emissions of about 4%. The air-sea exchange is mainly depending on the oceanic concentrations, the sea surface temperature and the wind speed.*

L256-257: "We only calculate bromoform source gas injection to the stratosphere". Do you mean that you do not consider product gases at all? Perhaps you should state this for clarity?

*This has been added.*

L268: what drives the high emissions along the NH coastlines? Macroalgae? I guess you imply this later on (L 280-281) but why not state it here first?

*Yes, we believe macroalgae to be the main reason for these elevated mixing ratios. We added it now.*

L272: what is meant by "elevated atmospheric mixing ratios"? Where would the elevated levels come from?

*During TransBrom three atmospheric regimes (Northern, Tropical and Southern Regime) could be identified. The Northern Regime from 42°N to 24°N was influenced by tropical storm activity during the cruise. In this regime, the air masses originated from East Russian and Japanese mainland and coastal areas. This air crossed the open ocean during the following days, where high biological productivity at the coast and open sea east of Japan was determined in-situ and via satellite measurements of chlorophyll-a (http://oceancolor.gsfc.nasa.gov/). Thus additional to the signature from oceanic phytoplankton, the atmosphere acquired anthropogenic, terrestrial and coastal properties, where higher amounts of CHBr3 are likely (e.g. Quack and Wallace, 2003; Ziska et al., 2013) and may have contributed to the elevations.*

L278: add "significant", i.e. "but show two significant differences. . ."

*Done.*

L279-283: If the Stemmler approach does not consider macroalgae and the effect of coastal processes then surely it will always underestimate bromoform emissions? How important are macroalgae relative to phytoplankton, particularly in these regions?

*This is a good question, which is hard to answer and definitely needs further research. Leedham et al. (2013) measured halocarbon fluxes from macro algae in Malaysia and provides a regional comparison to emissions derived from a top down approach by Pyle et al. (2011) for the South-East Asian region. There the largest estimate of the macro algal contribution was close to the lowest estimate from the entire oceanic region, revealing that the open ocean, respectively phytoplankton emissions appear larger, however it is also clear that the coastal emissions and those around macro algae can be >3 orders of magnitude higher than the open ocean fluxes. While the higher flux rates are confined to narrow coastal bands of unclear dimensions, the open ocean fluxes and hot spot emission therein extend over larger areas. Thus macro algal emissions appear higher when small regional scales are considered, and may lose significance when looking at larger ocean areas.*

*However the calculation of Ziska et al. (2013) claim that 70% of the bromoform fluxes in the first 200 km off the coasts. Here however, other sources like anthropogenic disinfection processes form coastal power plants or other industries and municipalities also contribute. While the Leedham and Ziska studies provided a good start for the investigation about the significance of coastal versus open ocean sources, both studies admit many uncertainties in their assumptions. Future research needs to address processes as well as sources to reveal the future development of halocarbon emissions.*

L287-289: It is not clear from Figure 2 that emissions are necessarily higher in winter and summer than they are in spring and fall. Can a more robust or statistical case be made (total flux from the region in each of the 4 seasons for example)?

*The annual cycle of emissions and the maximum emission seasons are already visible in Fig. 1. We added a reference to this figure.*

Figure 2: What is the cause of the high winter emissions (in the Ziska inventory) from the Chinese and Vietnamese (and Philippines?) coastlines? This appears to be a strong source region that you do not really discuss in the text. Given the prevailing NE winds at that time of year this could be an important source of bromoform to the tropics (see, for example, Ashfold et al., ACP. 15, 2015 or Oram et al., ACP, 17,2017)

*These high coastal emissions during DJF in the Ziska Updated inventory result from high oceanic concentrations of up to 40 pmolL$^{-1}$ along these coastlines and the high wind speeds especially during December and January (see the wind reanalysis plots in Fig. R2).*

[Figure]

**Figure R2: Ziska Updated atmospheric mixing ratio and oceanic concentration of CHBr3 (top row) and ERA-Interim monthly mean 10 m-wind speed above the Indian Ocean and West Pacific (bottom row).**

Line 317: I agree that the coastal emissions are similar in magnitude but they are vastly different in location.

> *We added "in magnitude" to the sentence.*

Table 1: If the numbers given are annual averages, what do the ranges shown represent?

> *The given range represents the spatial variation within the emission fields. We added this.*

Table 1: Are these numbers just for the Indian Ocean (i.e. not the full geographical area shown in Fig 2)? Please define what is meant by the Indian Ocean. Also I wonder if you should avoid using the term IO as it could be mistaken for iodine oxide!

> *We added the longitudinal range for the Indian Ocean values given here. We only use the abbreviation IO in figures and always explain them in the caption. In the text, we always spell it out to avoid this confusion.*

L352: add "flux", i.e. "To evaluate our flux and transport calculations"?

> Done.

L352-353: replace "available" with "selected", i.e. "from selected ship and aircraft campaigns".

> *Done.*

L356: should be "Table 3" not "Table 4"

> *Thank you for paying close attention!*

L368: Begin sentence with "It is likely that oceanic sources. ..."? Although please refer also to my general comments on Section 3.2 above.

*The sentence has been changed and the following has been added: "…and our modeled VMR are generally to low because we do not use global emissions."*

L382-384: another, and possibly more likely (?), explanation would be the underestimation of the role of convection in this region. How well does FLEXPART deal with convection?

*FLEXPART includes the Emanuel and Živkovic-Rothman (1999) convection scheme to resolve convective transport (Forster et al 2007). In our companion studies by Tegtmeier et al. (2013) and Fuhlbrügge et al. (2016) using also FLEXPART/ERA-Interim set-up, we showed how well this set up simulates observed aircraft measurements of VSLS in the UTLS during SHIVA and other tropical aircraft campaigns.*

L405: In this section I think you should describe Figs 4a and 4b before discussing Fig 4c. As written, it is a little confusing.

*We reorganized the paragraph accordingly, and hope that we clarified the meanings.*

L407-409: I am slightly confused by the term "transport efficiency" and how this was derived. In lines 258-260 it was defined slightly differently than it is here. How is the spatial distribution of transport efficiency independent of the emission scenario used when the mass emitted is different for the 2 scenarios? As I understand it, Fig 4c is a general picture which shows from which regions idealized particles will cross the CPT and has nothing to do with the bromoform emission inventories at all? If I am right, the term "bromoform delivery" in the Figure caption is misleading. A little clarification here would be appreciated.

*We adapted the definition of the "transport efficiency" in this paragraph, as it was slightly wrong. The spatial distribution of transport efficiency is not independent of the emission scenario used. Please see also our answer to Reviewer1.*
*It is true, that Fig. 4c shows where particles from the ocean will cross the CPT, but only for transport related to the lifetime of bromoform. That is why this transport efficiency distribution is only valid for bromoform delivery and why we name bromoform in the figure caption.*

Figure 4c: I am intrigued as to how the particles in the north east corner of the map get into the stratosphere during the summer months (JJA) when the prevailing winds in the region are from the southwest. Do they enter through the Indian monsoon or by some other mechanism?

*This transport is likely more related to the convective transport connected with the ITCZ. The Indian monsoon and the Asian monsoon anticyclone generally do not extend this far east above the West Pacific.*

L450: "Asian coastal areas" is a bit general. Which bit of Asia?

*Here we mean the tropical Asian coastlines; we changed the phrasing.*

L454: I think you need to define again what you mean by the "stratospheric entrainment region". Please explain clearly what is depicted in Figure 5 and how it differs from Figure 4. Does Figure 5 show the geographical location at the CPT where particles pass through to the stratosphere? If so, it seems odd that the southern tip of India is so important when I thought the main convection occurs further to the north?

*This figure really shows the geographical location at the CPT where bromoform passes through to the stratosphere. We changed the description in the text and figure caption. We believe that the tip of India is so important, because of a combination of high emissions and fast vertical transport. Although more trajectories cross the CPT farther north, this transport takes longer and thus most of the bromoform is already decayed.*

L470 (Section 3.4): this section would benefit from a better description of the difference between transport efficiency and entrainment (as discussed above).

*We tried to clarify this point in the revised manuscript. First, as noted above, we changed the term entrainment to "injection" and, second, we added another definition of the transport efficiency in order to remind of the difference: "The annual cycle of bromoform transport efficiency, which is the injection to the stratosphere divided by the total IO/WP emissions, displays two maxima, one in July and one in January (Fig. 6b)."*

L491-493: the temporal shift is not particularly obvious from Fig 6c.

*It is true; the seasonality shift is only one month and the maximum month remains July. Still, the three maximum months shift from MJJ to JJA. We changed the wording from "maximum injection season" to "maximum injection months".*

L497-500: This sentence is not very clear. How does the "differing annual cycles of bromoform entrainment to the stratosphere" influence the "regional pattern of entrainment to the stratosphere"?

*This is explained in the following paragraph (ll. 508-541), where we explain the spatial differences in stratospheric injection of bromoform caused by different temporal resolution of the emissions. This sentence is thought to give a transmission towards this paragraph.*

L552 and L553: exactly 50% higher or approximately 50% higher?

*Approximately. This has been added now.*

L556: What altitude range does the anticyclone typically cover?

*The anticyclone covers the upper troposphere and reaches into the lower stratosphere in a range between 8 and 18 km.*

L615-617: "seasonality is only affected by wind speed and sea surface pressure". Is that because the atmospheric and ocean concentrations are assumed to be constant throughout the year?

*Yes, correct! It has been added.*

L617-619: "The Indian Ocean has a pronounced seasonality in ocean currents and upwelling regions (Schott et al., 2009) affecting the biological productivity, surface bromoform concentrations, and emissions". Why include this sentence here? Do you mean to say that these are not included in the Ziska calculations? If so, please say so for clarity.

*Yes, done.*

L645-649: This last sentence is not clear. What contributes "approximately half of the total stratospheric VSLS-Br"? Source gases in general? Where does the other 50% come from – product gases?

*Yes, we changed it now.*

**Citations**

Bergman, J. W., Fierli, F., Jensen, E. J., Honomichl, S., and Pan, L. L.: Boundary layer sources for the Asian anticyclone: Regional contributions to a vertical conduit, Journal of Geophysical Research: Atmospheres, 118, 2560-2575, 10.1002/jgrd.50142, 2013.

Chen, B., Xu, X., Yang, S., and Zhao, T.: Climatological perspectives of air transport from atmospheric boundary layer to tropopause layer over Asian monsoon regions during boreal summer inferred from Lagrangian approach, Atmos. Chem. Phys. , 12, 5827-5839, 2012.

Fiehn, A., Quack, B., Hepach, H., Fuhlbrügge, S., Tegtmeier, S., Toohey, M., Atlas, E., and Krüger, K.: Delivery of halogenated very short-lived substances from the west Indian Ocean to the stratosphere during the Asian summer monsoon, Atmos. Chem. Phys., 17, 6723-6741, 10.5194/acp-17-6723-2017, 2017.

Fiehn, A., Quack, B., Marandino, C. A., and Krüger, K.: Transport Variability of Very Short Lived Substances From the West Indian Ocean to the Stratosphere, Journal of Geophysical Research: Atmospheres, 123, 10.1029/2017JD027563, 2018.

Fuhlbrügge, S., Quack, B., Tegtmeier, S., Atlas, E., Hepach, H., Shi, Q., Raimund, S., and Krüger, K.: The contribution of oceanic halocarbons to marine and free tropospheric air over the tropical West Pacific, Atmos. Chem. Phys. , 16, 7569-7585, 10.5194/acp-16-7569-2016, 2016.

Gottschaldt, K. D., Schlager, H., Baumann, R., Bozem, H., Eyring, V., Hoor, P., Jöckel, P., Jurkat, T., Voigt, C., Zahn, A., and Ziereis, H.: Trace gas composition in the Asian summer monsoon anticyclone: a case study based on aircraft observations and model simulations, Atmos. Chem. Phys., 17, 6091-6111, 10.5194/acp-17-6091-2017, 2017.

Hossaini, R., Chipperfield, M. P., Feng, W., Breider, T. J., Atlas, E., Montzka, S. A., Miller, B. R., Moore, F., and Elkins, J.: The contribution of natural and anthropogenic very short-lived species to stratospheric bromine, Atmos. Chem. Phys. , 12, 371-380, 10.5194/acp-12-371-2012, 2012.

Hossaini, R., Patra, P. K., Leeson, A. A., Krysztofiak, G., Abraham, N. L., Andrews, S. J., Archibald, A. T., Aschmann, J., Atlas, E. L., Belikov, D. A., Bönisch, H., Carpenter, L. J., Dhomse, S., Dorf, M., Engel, A., Feng, W., Fuhlbrügge, S., Griffiths, P. T., Harris, N. R. P., Hommel, R., Keber, T., Krüger, K., Lennartz, S. T., Maksyutov, S., Mantle, H., Mills, G. P., Miller, B., Montzka, S. A., Moore, F., Navarro, M. A., Oram, D. E., Pfeilsticker, K., Pyle, J. A., Quack, B., Robinson, A. D., Saikawa, E., Saiz-Lopez, A., Sala, S., Sinnhuber, B. M., Taguchi, S., Tegtmeier, S., Lidster, R. T., Wilson, C., and Ziska, F.: A multi-model intercomparison of halogenated very short-lived substances (TransCom-VSLS): linking oceanic emissions and tropospheric transport for a reconciled estimate of the stratospheric source gas injection of bromine, Atmos. Chem. Phys. , 16, 9163-9187, 10.5194/acp-16-9163-2016, 2016.

James, R., Bonazzola, M., Legras, B., Surbled, K., and Fueglistaler, S.: Water vapor transport and dehydration above convective outflow during Asian monsoon, Geophysical Research Letters, 35, L20810, 10.1029/2008gl035441, 2008.

Leedham, E. C., Hughes, C., Keng, F. S. L., Phang, S. M., Malin, G., and Sturges, W. T.: Emission of atmospherically significant halocarbons by naturally occurring and farmed tropical macroalgae, Biogeosciences, 10, 3615-3633, 10.5194/bg-10-3615-2013, 2013.

Liang, Q., Stolarski, R. S., Kawa, S. R., Nielsen, J. E., Douglass, A. R., Rodriguez, J. M., Blake, D. R., Atlas, E., and Orr, L. E.: Finding the missing stratospheric Bry: a global modeling study of CHBr3 and CH2Br2, Atmos. Chem. Phys. , 2269-2286, 2010.

Liang, Q., Atlas, E., Blake, D. R., Dorf, M., Pfeilsticker, K., and Schauffler, S.: Convective transport of very short lived bromocarbons to the stratosphere, Atmos. Chem. Phys. , 14, 5781-5792, 2014.

Orbe, C., Waugh, D. W., and Newman, P. A.: Air-mass origin in the tropical lower stratosphere: The influence of Asian boundary layer air, Geophysical Research Letters, 42, 4240-4248, 10.1002/2015gl063937, 2015.

Pan, L. L., Honomichl, S. B., Kinnison, D. E., Abalos, M., Randel, W. J., Bergman, J. W., and Bian, J.: Transport of chemical tracers from the boundary layer to stratosphere associated with the dynamics of the Asian summer monsoon, Journal of Geophysical Research: Atmospheres, 121, 14,159-114,174, 10.1002/2016jd025616, 2016.

Park, M., Randel, W. J., Emmons, L. K., and Livesey, N. J.: Transport pathways of carbon monoxide in the Asian summer monsoon diagnosed from Model of Ozone and Related Tracers (MOZART), Journal of Geophysical Research: Atmospheres, 114, D08303, 10.1029/2008JD010621, 2009.

Ploeger, F., Konopka, P., Müller, R., Fueglistaler, S., Schmidt, T., Manners, J. C., Grooß, J. U., Günther, G., Forster, P. M., and Riese, M.: Horizontal transport affecting trace gas seasonality in the Tropical Tropopause Layer (TTL), Journal of Geophysical Research: Atmospheres, 117, D09303, 10.1029/2011jd017267, 2012.

Pyle, J. A., Ashfold, M. J., Harris, N. R. P., Robinson, A. D., Warwick, N. J., Carver, G. D., Gostlow, B., O'Brien, L. M., Manning, A. J., Phang, S. M., Yong, S. E., Leong, K. P., Ung, E. H., and Ong, S.: Bromoform in the tropical boundary layer of the Maritime Continent during OP3, Atmos. Chem. Phys. , 11, 529-542, 10.5194/acp-11-529-2011, 2011.

Quack, B., and Wallace, D. W. R.: Air-sea flux of bromoform: Controls, rates, and implications, Global Biogeochemical Cycles, 17, 1023, 10.1029/2002gb001890, 2003.

Randel, W. J., Park, M., Emmons, L., Kinnison, D., Bernath, P., Walker, K. A., Boone, C., and Pumphrey, H.: Asian monsoon transport of pollution to the stratosphere, Science, 328, 611-613, 10.1126/science.1182274, 2010.

Russo, M. R., Ashfold, M. J., Harris, N. R. P., and Pyle, J. A.: On the emissions and transport of bromoform: sensitivity to model resolution and emission location, Atmos. Chem. Phys., 15, 14031-14040, 10.5194/acp-15-14031-2015, 2015.

Santee, M. L., Manney, G. L., Livesey, N. J., Schwartz, M. J., Neu, J. L., and Read, W. G.: A comprehensive overview of the climatological composition of the Asian summer monsoon anticyclone based on 10 years of Aura Microwave Limb Sounder measurements, Journal of Geophysical Research: Atmospheres, 122, 5491-5514, 10.1002/2016jd026408, 2017.

Stemmler, I., Hense, I., and Quack, B.: Marine sources of bromoform in the global open ocean - global patterns and emissions, Biogeosciences, 12, 1967-1981, 10.5194/bg-12-1967-2015, 2015.

Tegtmeier, S., Krüger, K., Quack, B., Atlas, E., Blake, D. R., Boenisch, H., Engel, A., Hepach, H., Hossaini, R., Navarro, M. A., Raimund, S., Sala, S., Shi, Q., and Ziska, F.: The contribution of oceanic methyl iodide to stratospheric iodine, Atmos. Chem. Phys. , 13, 11869-11886, 10.5194/acp-13-11869-2013, 2013.

Tissier, A. S., and Legras, B.: Convective sources of trajectories traversing the tropical tropopause layer, Atmos. Chem. Phys. , 16, 3383-3398, 10.5194/acp-16-3383-2016, 2016.

Vogel, B., Günther, G., Müller, R., Grooß, J. U., and Riese, M.: Impact of different Asian source regions on the composition of the Asian monsoon anticyclone and on the extratropical lowermost stratosphere, Atmos. Chem. Phys. , 15, 13699-13716, 10.5194/acpd-15-13699-2015, 2015.

Vogel, B., Günther, G., Müller, R., Grooß, J. U., Afchine, A., Bozem, H., Hoor, P., Krämer, M., Müller, S., Riese, M., Rolf, C., Spelten, N., Stiller, G. P., Ungermann, J., and Zahn, A.: Long-range transport pathways of tropospheric source gases originating in Asia into the northern lower stratosphere during the Asian monsoon season 2012, Atmos. Chem. Phys. , 16, 15301-15325, 10.5194/acp-16-15301-2016, 2016.

Wales, P. A., Salawitch, R. J., Nicely, J. M., Anderson, D. C., Canty, T. P., Baidar, S., Dix, B., Koenig, T. K., Volkamer, R., Chen, D., Huey, L. G., Tanner, D. J., Cuevas, C. A., Fernandez, R. P., Kinnison, D. E., Lamarque, J.-F., Saiz-Lopez, A., Atlas, E. L., Hall, S. R., Navarro, M. A., Pan, L. L., Schauffler, S. M., Stell, M., Tilmes, S., Ullmann, K., Weinheimer, A. J., Akiyoshi, H., Chipperfield, M. P., Deushi, M., Dhomse, S. S., Feng, W., Graf, P., Hossaini, R., Jöckel, P., Mancini, E., Michou, M., Morgenstern, O., Oman, L. D., Pitari, G., Plummer, D. A., Revell, L. E.,

Rozanov, E., Saint-Martin, D., Schofield, R., Stenke, A., Stone, K. A., Visioni, D., Yamashita, Y., and Zeng, G.: Stratospheric Injection of Brominated Very Short-Lived Substances: Aircraft Observations in the Western Pacific and Representation in Global Models, Journal of Geophysical Research: Atmospheres, 123, 10.1029/2017JD027978, 2018.

Yamamoto, H., Yokouchi, Y., Otsuki, A., and Itoh, H.: Depth profiles of volatile halogenated hydrocarbons in seawater in the Bay of Bengal, Chemosphere, 45, 371-377, 10.1016/S0045-6535(00)00541-5, 2001.

Yan, R., and Bian, J.: Tracing the boundary layer sources of carbon monoxide in the Asian summer monsoon anticyclone using WRF-Chem, Adv. Atmos. Sci., 32, 943-951, 10.1007/s00376-014-4130-3, 2015.

Ziska, F., Quack, B., Abrahamsson, K., Archer, S. D., Atlas, E., Bell, T., Butler, J. H., Carpenter, L. J., Jones, C. E., Harris, N. R. P., Hepach, H., Heumann, K. G., Hughes, C., Kuss, J., Krüger, K., Liss, P., Moore, R. M., Orlikowska, A., Raimund, S., Reeves, C. E., Reifenhäuser, W., Robinson, A. D., Schall, C., Tanhua, T., Tegtmeier, S., Turner, S., Wang, L., Wallace, D., Williams, J., Yamamoto, H., Yvon-Lewis, S., and Yokouchi, Y.: Global sea-to-air flux climatology for bromoform, dibromomethane and methyl iodide, Atmos. Chem. Phys. , 13, 8915-8934, 10.5194/acp-13-8915-2013, 2013.